# Fast and Accurate Spreading Process Temporal Scale Estimation

**Abram Magner** *amagner@albany.edu*
*University at Albany, SUNY*

**Carolyn Kaminski** *ckaminski@albany.edu*
*University at Albany, SUNY*

**Petko Bogdanov** *pbogdanov@albany.edu*
*University at Albany, SUNY*

**Reviewed on OpenReview:** *https://openreview.net/forum?id=k4iWTEdUSF*

## Abstract

Spreading processes on graphs arise in a host of application domains, from the study of online social networks to viral marketing to epidemiology. Various discrete-time probabilistic models for spreading processes have been proposed. These are used for downstream statistical estimation and prediction problems, often involving messages or other information that is transmitted along with infections caused by the process. These models generally model cascade behavior at a small time scale but are insufficiently flexible to model cascades that exhibit intermittent behavior governed by multiple scales. We argue that the presence of such time scales that are unaccounted for by a cascade model can result in degradation of performance of models on downstream statistical and time-sensitive optimization tasks. To address these issues, we formulate a model that incorporates multiple temporal scales of cascade behavior. This model is parameterized by a *clock*, which encodes the times at which sessions of cascade activity start. These sessions are themselves governed by a small-scale cascade model, such as the discretized independent cascade (IC) model. Estimation of the multiscale cascade model parameters leads to the problem of *clock estimation* in terms of a natural distortion measure that we formulate. Our framework is inspired by the optimization problem posed by DiTursi et al, 2017, which can be seen as providing one possible estimator (a maximum-proxy-likelihood estimator) for the parameters of our generative model. We give a clock estimation algorithm, which we call FastClock, that runs in linear time in the size of its input and is provably statistically accurate for a broad range of model parameters when cascades are generated from any spreading process model with well-concentrated session infection set sizes and when the underlying graph is at least in the semi-sparse regime. We exemplify our algorithm for the case where the small-scale model is the discretized independent cascade process and extend substantially to processes whose infection set sizes satisfy a general martingale difference property. We further evaluate the performance of FastClock empirically in comparison to the state of the art estimator from DiTursi et al, 2017. We find that in a broad parameter range on synthetic networks and on a real network, our algorithm substantially outperforms that algorithm in terms of both running time and accuracy. In all cases, our algorithm's running time is asymptotically lower than that of the baseline.

## 1 Introduction

There are a variety of well-established and simple probabilistic generative models for graphs and and infectious processes that run over these graphs. In this work we specifically focus on models for spreading

processes on networks such as the diffusion of innovation (Montanari & Saberi, 2010), information (Bakshy et al., 2012) and misinformation (Shin et al., 2018) in social networks. In such models, every node at a given time has a *state*, and the model stipulates the conditional probabilities of nodes being in given states at future times, given their states at the current time.

Typically, these models only focus on a single, uniform time scale, thus ignoring global synchronizing events that occur in reality (such as periodic daily/weekly business schedules governing the actions of network agents). In this work, we address modeling and estimation of parameters in the common scenario where spreading process activity occurs intermittently, according to a global synchronizing schedule, as in Example 1.

**Example 1** (A scenario leading to intermittent spreading process activity)**.** *Consider a cascade (of, say, a hashtag) among users of an online social network. These users may interact with the network only during certain times of day, owing to their work schedules. For instance, the bulk of interactions may occur in the morning, during one's lunch break, during one's afternoon tea, and after work. Infections are likely to proceed only during these periods of activity, during which they spread stochastically according to a more granular temporal process. See Example 2 for a refinement of this scenario, using the terminology of our model.*

The scenario just presented involves two time scales:

1. A larger time scale, consisting of a sequence of *sessions* during which local spreading process activity occurs.

2. A smaller time scale during each session, consisting of local spreading process activity as captured by a typical model.

It is important to take these multiple scales into account in model building and subsequent parameter estimation, for a variety of reasons:

1. Failing to take the larger time scale into account (i.e., assuming that the process is entirely governed by the smaller time scale) leads to erroneous parameter estimates for the smaller time scale model.

2. Even when the smaller time scale model parameters are known, failure to account for the larger time scale leads to error in downstream statistical inference and optimization tasks, such as cascade doubling time prediction (Cheng et al., 2014) and message model parameter estimation.

3. An estimate of the larger time scale or parameters of a model generating it can themselves be used as features in machine learning problems involving cascades, such as cascade classification (e.g., where class labels are "misinformation" or "true information").

We discuss a few motivations in more detail in Section 1.2. Furthermore, in general, given cascade data in a large network, the form of the intermittencies in the large time scale are not obvious and may vary depending on the community from which the cascades originated since different communities and cascades consisting of different types of information (e.g., long-form essays versus infographics versus videos versus single-panel web comics) may operate according to different schedules. It is thus important to learn these intermittencies from observations, as we do in the present work.

Given these motivations, the present paper is concerned with

- the formulation of a generative model for network cascades that is parametrized by an arbitrary *large* time scale (which itself may contain events on multiple scales), with the smaller time scale generated by some given standard model;

- the formulation of the estimation problem for the parameters of the large time scale from cascade observations;

- algorithmic techniques for provably accurate and computationally efficient solution of the above estimation problem.

Our work is inspired by the works of DiTursi et al. (2017) and DiTursi et al. (2019), which took the following approach: they formulated a problem of *clock recovery*, in which a "ground-truth" cascade is generated by the *independent cascade model* of Kempe et al. (2003), then *perturbed* by a "clock transformation", parametrized by a clock (a sequence of observation timestep endpoints) to yield an *observed* cascade. The task that they formulated is an optimization one: from the observed cascade, recover the clock with maximum likelihood. Among other contributions, they formulated a dynamic programming algorithm that provably achieves this maximum likelihood, but with a large asymptotic running time (see the discussion in Section 1.3). **Concretely, their dynamic programming algorithm can be seen as implementing one possible estimator for the clock parameter of our generative model, in the case where the cascade, viewed on the large time scale, is distributed according to the independent cascade model of Kempe et al. (2003).** We are the first to empirically study the *accuracy* of their estimator, in terms of distance to the ground truth parameters, and, thus, it is not a priori clear what types of graphs are most favorable for its performance.

In the present work, we take a more statistical approach: we rephrase the clock estimation problem via a generative model that produces cascades with multiple temporal scales. In this model, clocks become the *parameters* of the larger temporal scale, and the task is to *estimate* the unknown clock (the parameter) from the data that the model produces. We make this estimation task rigorous by introducing a notion of distortion between clocks, which measures the accuracy of an estimated clock. The clocks in our work are the same type of object as those in the prior work (but interpreted differently).

Our reformulation is valuable for both performance and philosophical reasons:

1. In the prior work, it is unclear what physical mechanism is being modeled by the clock transformation which operates on an already generated cascade, while, in contrast, in our work, clocks are a natural product of global intermittent influences on individual node behavior.

2. In the prior work, the "ground-truth" cascade uses a notion of timesteps that do not map to physical times, while in our work, the observed cascade operates in a discretization of real, physical time.

3. Prior work does not attempt, even empirically, to quantify the distance between the ground-truth clock (the one that actually generated the data) and the estimated one – maximizing the likelihood of the estimated clock, rather than ensuring accuracy, is treated as the end goal. In contrast, in our work we give a rigorous estimation formulation, which allows us to quantify the accuracy of the maximum likelihood estimator versus alternatives (such as the estimator that we introduce). As detailed throughout this paper, our resulting estimator substantially outperforms the previous work's dynamic programming-based maximum likelihood estimator in terms of running time and, on all of our synthetic data and in a broad range of parameters on real networks, in terms of accuracy. Across the entire parameter range on real networks, the average accuracy of our estimator is quite low (consistently less than 0.07).

## 1.1 High-level problem formulation

We next state our problem at a slightly more technical level (we give all formal formulations in Section 2). To do so, we first introduce preliminaries on spreading process models.

**Preliminaries: spreading process models.** Several well-studied information diffusion models on a graph $G$ assume a discrete timeline in which at every time step nodes of $G$ participate in the diffusion process (i.e., become "infected") based on influence from network neighbors who became infected in past time steps. Here, a timeline is simply an ordered sequence of indices: $0, 1, 2, \ldots$. The output of a spreading process model after timestep $N$ is an *infection sequence* of disjoint vertex subsets $S_0, S_1, ..., S_N$, where $S_t$ is the set of vertices of $G$ that became infected during timestep $t$. For example, according to the *independent cascade model* infected nodes have one chance to infect their neighbors, while in the *linear threshold model* nodes

get infected when a critical fraction of their neighbors have been infected in any prior time steps (Kempe et al., 2003). In addition to the discrete time models above, there also exist continuous-time counterparts. For the purposes of computation, the latter are generally discretized by a standard recipe: given a resolution $\delta > 0$ and a cascade sampled from a continuous-time model over a time interval $[0, T]$, one defines the $\delta$-discretization of the cascade to be the infection sequence $S = (S_0, S_1, ..., S_{\lceil T/\delta \rceil})$, where $S_j$ consists of the set of vertices infected during the time interval $j\delta$. In the discussion that follows, we will thus state our formulation in the discrete-time case, with the understanding that this is without loss of generality.

**Formulation of our two-scale model.** To capture our motivating scenario, we will define a new two-scale cascade model, where the smaller scale is given by an arbitrary fixed spreading process model $M_0$ with the property that, at any given time $t$, a vertex may only transmit infection if it is in the set of *active nodes* $A_t$ at that time.

The generative process that we will define is parametrized by a sequence of time points $t_0, t_1, ..., t_N$, giving the larger time scale. Intuitively, $t_j$ is the time at which the $j$th session starts. The activity within the $j$th session (comprising the *smaller scale* of the model) is dictated by $M_0$ as follows: for any time $t \in [t_j, t_{j+1})$, the distribution of new vertex infection events is given by $M_0$ conditioned on the active set remaining fixed to its value at time $t_j$. In other words, vertices may become *infected* in the time interval $[t_j, t_{j+1})$, but the active set is not updated until time $t_{j+1}$. This assumption is justified whenever nodes' "attention spans" have a finite granularity, in the sense that newly infected vertices do not become infectious immediately. This is aligned with the intuition of daily activity periods from Example 1 and more broadly justified in spreading processes on social networks. We give a more specific scenario in Example 2 to illustrate this. In the appendix, Section G, we explain the behavior of our estimator when this assumption does not strictly hold.

**Example 2** (Scenarios illustrating our active set intermittent update assumption)**.** *Here, we give more detailed examples of scenarios in which spreading process activity satisfies our assumption regarding intermittent updating of the active set.*

*Consider a social network such as Facebook, where nodes are people and edges are friendships. We consider cascades in which a person is said to be infected when they post a particular piece of content. Note that infection of a person is detectable immediately, and can thus be recorded in a dataset. A person is active (i.e., infectious) when other people see what that person has posted. We assume, furthermore, that people have, in any given session (defined below), a limited reserve of attention, and will thus be focused on posts that occurred prior to the current session.*

*Now, a session starts when some break in the day happens and ends when the break stops. At the start of a cascade, some initially active vertices $S_{sess}(0)$ are seen by users in the first session. A subset $S_{sess}(1)$ of these users, during that same session, repost, becoming infected. Neighbors of newly infected vertices do not see these reposts during this session, because of the aforementioned limitations of attention. Thus, the elements of $S_{sess}(1)$ are infected, but they are not active until the* next *session (which may occur several hours after the current one), at the beginning of which the attention reserves of their neighbors have been replenished.*

*An alternative mechanism that could produce the same effect as the attention reserve is a platform playing an active role in displaying posts non-chronologically, resulting in delays in the display of a post.*

*We note that in the above example, different types of content may plausibly spread according to different clocks. For instance, long videos may be shared only after work hours, while technical essays may be shared during work. Comedic webcomics may be shared throughout the day. Thus, it is of interest to estimate the clocks of different cascades from data, rather than assuming some hard-coded clocks.*

**Formulation of our estimation problem.** Having formulated our two-scale model, we next formulate the central estimation question that we study in this work: given a sample infection sequence $(S_{obs}(0), S_{obs}(1), ..., S_{obs}(\hat{N}))$ from the two-scale model parametrized by the larger time scale $t_0, t_1, ..., t_N$, the natural question to ask is to what extent $t_0, ..., t_N$ can be estimated. For many models, this is reducible to a more fundamental problem, which is the central one that we tackle in this work: estimate the assignment of

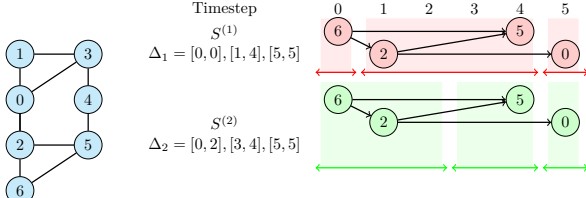

Figure 1: An example network and a cascade $S_{obs}$ with corresponding observed infection sequence $S_{obs} = (\{6\}, \{2\}, \{\}, \{\}, \{5\}, \{0\})$, encoding that 6 was infected in timestep 0, 2 in timestep 1, 5 in timestep 4, and 0 in timestep 5. Suppose that this cascade was generated according to our two-scale model with clock $\Delta_1 = ([0, 0], [1, 4], [5, 5])$, so that there are 6 timesteps and 3 sessions. Then the (unobserved) session-level infection sequence is $S_{sess}^{(1)} = (\{6\}, \{2, 5\}, \{0\})$. An estimator of the clock, given $S_{obs}$, might output a different clock: say, $\Delta_2 = ([0, 2], [3, 4], [5, 5])$, yielding an estimated session-level infection sequence of $S_{sess}^{(2)} = (\{6, 2\}, \{5\}, \{0\})$. In Definition 4, we will define a distance between clocks, under which $\Delta_1$ and $\Delta_2$ have nonzero distance, because $\Delta_2$ entails that vertices 6 and 2 were both infected in the first session.

vertex infection events to the indices of sessions during which they occurred. We call this the *clock estimation problem*. We elucidate this estimation problem by the example in Figure 1.

## 1.2 Motivating applications

We next discuss several motivating applications for our generative model and for the clock recovery problem. In the listed applications, we explain how failing to account for multiple time scales results in degraded statistical accuracy or optimization performance on downstream tasks. We note that all of these applications involve a nontrivial statistical or optimization task even when the parameters of the small-scale cascade model are known.

1. **Parameter estimation for information models running on top of cascade processes:** In one class of applications, one wants to make inferences about some extra piece of information (a message, sentiment, opinion, etc.) that is spread along with an infection, according to a parametric statistical model (which we will call, for this discussion, an information model), and the information associated with a given node is statistically dependent on the corresponding information associated with the nodes that infected it.

   As a concrete example, consider a model of the spread of a sentiment (an information model) that runs on cascades generated by our two-scale model parametrized by some nontrivial clock, with the smaller scale generated by the discretized independent cascade model. In particular, the seed nodes $v \in S_0$ of the cascade are endowed with sentiments $X_v$ uniformly distributed in $[0, 1]$. When a node $w$ becomes infected, say, by an unknown subset $I(w)$ of active neighbors of $w$, the sentiment of $w$ is given as follows:

   $$X_w = \theta_1 \theta_2 + (1 - \theta_2) \frac{1}{|I(w)|} \sum_{v \in I(W)} X_v. \tag{1}$$

   Here, $\theta_1$ is a bias parameter $\in [0, 1]$ that represents some base sentiment, and $\theta_2 \in [0, 1]$ is a social influence parameter that governs how much each node $w$ tends to adopt the average sentiment of its infectors. The goal in such a model is to estimate $\theta_1, \theta_2$ from observation of a cascade as well as measured sentiments $X_w$.

   Note that even when the parameters of the smaller-scale spreading process model are known, it is of interest to estimate the parameters of the information model from sample cascades. Furthermore, in such applications, clock estimation – the main subject of this paper – is an essential preprocessing step in order to reduce the complexity of parameter estimation by determining the set of vertices that *could have* infected a given vertex $w$ (i.e., the set of active neighbors of $w$). This set is not observable, owing to lack of knowledge of the clock. Furthermore, failure to consider multiple time scales introduces error into the log likelihood function of the information model parameters since

the correct set of active neighbors is unknown. Ultimately, this results in erroneous parameter estimation.

2. **Time-critical tasks:** In certain applications, the goal is to optimize some objective function defined on the state of the process, subject to a time deadline that is given as input. For instance, in time-critical influence maximization (Chen et al., 2012b), the problem is to choose seed nodes to optimize the expected number of infections occurring by a given timestep.

   For a given seed set, the expected number of vertices infected by the deadline may change dramatically depending on whether or not a large time scale is incorporated into the model. Intuitively, this is because delays introduced by a large time scale may make the infection of a particularly well-connected vertex less likely. Thus, taking into account multiple time scales is essential for the success of model-based time-critical optimization tasks. This is exacerbated even further in cases where different subsets of nodes operate according to different large time scales.

3. **Cascade doubling time prediction:** The cascade doubling time prediction problem was posed in Cheng et al. (2014). A cascade is observed up to a certain time $t$, and the goal is to produce an accurate estimate of the time at which the number of infected nodes doubles from the number at time $t$. This problem was introduced as a prototypical prediction problem for cascades. In Example 3 in Appendix C, we spell out the details of a concrete example in which failure to account for a large time scale results in avoidable inaccuracy in doubling time prediction.

### 1.3 Prior work

The general topic of analysis of cascades has received a large amount of attention, both from theoretical and empirical perspectives. There are many cascade models, with features depending on application domains. For example, the independent cascade (IC) and linear threshold (LT) models were popularized in Kempe et al. (2003) for the application of *influence maximization*. This problem continues to be studied, even in settings where cascade model parameters are known (Lee et al., 2016; Abbe et al., 2017). Variations on the influence maximization problem that have time-critical components and, thus, require accurate modeling of multiple time scales in the sense that we study here, have also been studied (Chen et al., 2012a; Ali et al., 2019). These models are also used outside the context of influence maximization, e.g., in modeling the spread of memes on social networks (Adamic et al., 2016).

Statistical prediction tasks involving cascades have also been posed. For instance, the cascade doubling time prediction task was considered in Cheng et al. (2014). Other works propose models in which a piece of information, such as a message, an opinion, or a viral genome, is transmitted along with the infection of a node (Eletreby et al., 2020; De et al., 2016; Park et al., 2020). For such statistical problems, statistical inferences about the transmitted information can be disrupted by inaccurate estimation of the set of infectious vertices at a given time, further motivating generative modeling and estimation that takes into account multiple time scales.

The estimation method that we propose and study here, *FastClock*, bears a resemblance to methods in the *online change-point detection* literature (Veeravalli & Banerjee, 2014). Broadly speaking, the goal of that problem is to sequentially observe independent and identically distributed random variables $X_1, X_2, ...$ and to detect, with as few samples as possible, an index after which the distribution of the variables changes. One procedure for this, called *CuSum*, evaluates for each index $j$ a statistic $Y_j = Y_j(X_j)$ and maintains a sum $Z_t = \sum_{j=1}^{t} Y_t$. A changepoint is declared once $Z_t$ exceeds a certain threshold. In this paper, we define a similar rule for detecting *session endpoints*. In the context of change-point detection, CuSum has been analyzed in both the iid and more general hidden Markov setting (Fuh, 2003). However, our analysis is necessarily substantially different from that of CuSum, which rests on stringent assumptions (e.g., that the $X_j$ are sampled from an ergodic Markov chain).

In DiTursi et al. (2017) (see also followup work in DiTursi et al. (2019)), the authors formulated a version of the problem of clock recovery from cascade data generated according to an adversarially chosen clock as a problem of maximization of a function of the clock that serves as a proxy (in particular, an upper bound) for the log likelihood of the observed cascades. They proposed a solution to this problem via a

dynamic programming algorithm. While the dynamic programming algorithm is an exact solution to their formulation of the problem, it has a running time of $\Theta(n^4)$, where $n$ is the total number of vertices in the graph on which the observed cascade runs. This is prohibitively expensive for graphs of moderate to large size. Furthermore, their formulation of the problem makes no comparison of the estimated clock with the ground truth one, and thus there are no theoretical guarantees or empirical evaluations of the accuracy of their estimator (which we call the *maximum likelihood proxy (MLP) estimator*) as an approximation to the ground truth clock. In contrast, the present work gives a rigorous formulation of the problem as one of statistical estimation of the ground truth clock from observed cascades. We compare our proposed algorithm and estimator with the MLP estimator in this framework in terms of both accuracy and running time.

### 1.4 Our contributions

Our contributions in this work are as follows:

• **Novel problem formulation.** We formulate a two-scale generative model for cascades, where the larger time scale is parametrized by an arbitrary clock (which may itself be generated by a model with arbitrarily many time scales). We formalize the problem of clock estimation from observed cascades with respect to a natural distortion measure. This distortion measure allows us to quantify the proximity of estimated clocks to the ground truth in a principled manner.

• **Provably accurate and computationally efficient solutions.** We propose a linear-time algorithm with provable approximation guarantees for the clock estimation problem. In particular, the distortion between the true clock and our estimate tends to 0 at a polynomial rate with respect to the average degree of the graph.

• **Generality.** We first prove our results in the context of Erdős-Rényi graphs in the semi-sparse regime and the independent cascade model. The proofs for this setting contain almost all necessary ingredients for extension of guarantees to a much more general setting: the results also hold when the cardinalities of infection sets $S_i$ satisfy a martingale difference property and the graph is sampled from a sufficiently dense *sparse graphon* model. We provide the remaining ingredients in Section 3.5 and culminate with a more general theorem.

• **Confirmation of results in simulation on synthetic and real graphs.** We bolster our theoretical results via experiments on both real and synthetic graphs and synthetic cascades. We find that the FastClock estimator empirically outperforms the dynamic programming-based estimator from DiTursi et al. (2017) in terms of accuracy and on all of our synthetic graphs and in a broad range of cascade model parameters on a real graph, and substantially in terms of running time on all of our synthetic graphs and for all investigated model parameters on a real graph. We give our simulation results in Section 3.4 and Appendix D.

## 2 Problem formulation and notation

Our goal in this section is to formulate our generative model and the problem of *clock estimation*. We give examples of all definitions in Appendix C.

**Preliminary definitions:** We fix a graph $G$ on the vertex set $[n] = \{1, ..., n\}$, and we define the *timeline of length $N$*, for any number $N \in \mathbb{N}$, to be the set $[[N]] = \{0, 1, ..., N\}$. The first ingredient of our framework is a *cascade model*. We will be concerned in the present work with compartmental models, wherein each node may be susceptible (uninfected), active (i.e., infected and able to transmit the infection to its neighbors), or infected (but unable to transmit its infection). We collect the different possible compartments into a set $\Omega = \{S, A, I\}$. Nodes may carry additional information from an arbitrary set $\Omega'$. At any given time $t$, a node $v \in [n]$ is in some state $\Psi(v, t) \in \Omega \times \Omega'$. We call a *network state* any element of $(\Omega \times \Omega')^n$.

**Definition 1** (Cascade model, observation model). *A (discrete-time) cascade model $\mathcal{C}$ is a conditional distribution $P_{\mathcal{C}}(\cdot \mid X)$, where $X \in (\Omega \times \Omega')^n$ is a network state.*

*For the bulk of the paper, we will be concerned with the following model of observations of cascades: for a sequence of network states $(X_0, X_1, ..., X_N)$, we observe a corresponding sequence, called an* infection

sequence: $(S_{obs}(0), S_{obs}(1), ..., S_{obs}(N))$, where $S_{obs}(j)$ consists of the set of vertices at time $j$ that move into either compartment $A$ or $I$ according to $X_j$. Note that we do not observe which vertices are active or any of the side information. Any cascade model thus induces a probability distribution on infection sequences.

To begin to define our two-scale generative model, we next define a *clock*, which encodes the timesteps belonging to each session. This will be a parameter of our model.

**Definition 2** (Clock). *A clock $C$ with $N+1$ sessions on the timeline $[[N']]$ is a partition of $[[N']]$ into $N+1$ closed subintervals (i.e., contiguous integer subsets) $C_0, ..., C_N$. We call the $j$th such subinterval, for $j = 0$ to $N$, the $j$th session interval.*

*Given an observed infection sequence $S_{obs} = (S_{obs}(0), ... S_{obs}(N'))$, a clock $C$ induces an infection sequence $S_{sess} = (S_{sess}(0), S_{sess}(1), ..., S_{sess}(N))$, where $S_{sess}(j) = \cup_{k \in C_j} S_{obs}(k)$. We call this the infection sequence induced by $C$ on $S_{obs}$.*

For algorithmic purposes, we note that a clock may be encoded as a sequence of non-negative integers giving its right interval endpoints.

**Main definition of the generative model:** We now define our main generative model. We fix a cascade model $\mathcal{C}_0$, which will govern the dynamics of the process during each session. This is the *small-scale* model. In general, it may depend on the length of the session.

**Definition 3** (Two-scale generative model for cascades). *The two-scale generative model $\mathcal{M}(C, X_0, T) = \mathcal{M}(C)$ is parametrized by a clock $C$, an initial network state $Z_0 \in (\Omega \times \Omega')^n$, and a number of sessions $T$. Its output is a sequence $(X_0, X_1, ..., X_N)$ of network states, inducing an observed infection sequence $(S_{obs}(0), ..., S_{obs}(N))$.*

*For $j = 1, ..., T$, at the beginning of the $j$th session, the network state is given by $Z_{j-1}$. At each timestep $t$ in the $j$th session, a network state is sampled from $\mathcal{C}_0$, conditioned on the current network state. Any vertex that is newly activated according to $\mathcal{C}_0$ is set to infected in $X_t$ and active in $Z_j$. For convenience, we also define the infection sequence $S_{sess} = (S_{sess}(0), S_{sess}(1), ..., S_{sess}(T))$, where $S_{sess}(j)$ consists of the set of vertices infected during session $j$ (so $S_{sess}$ is just the infection sequence induced by $C$ on $S_{obs}$). We call $S_{sess}$ the session-level infection sequence.*

Intuitively, in this model, vertices infected during a given session only become *active* in the next session. This is justified in scenarios where nodes are not immediately infectious when they become infected. In this case, sessions may be seen as periods during which a current set of active vertices causes infections.

**Measuring distortion between clocks:** To formulate the problem of estimating the clock of our model, given an infection sequence produced by it, we need a means of measuring distortion between clocks. This is our next goal.

An infection sequence $S$ naturally induces a partial order on the set of vertices: namely, for two vertices $a, b$, $a < b$ if and only if $a \in S_i, b \in S_j$ for some $i < j$. Similarly, a clock $C$ applied to an infection sequence $S_{obs}$, in the sense of Definition 2, induces a partial order. This partial order is the one induced by the infection sequence $S$ that $C$ induces on $\hat{S}$.

We will consider two clocks $C_0, C_1$ to be equivalent with respect to a given observed infection sequence $S_{obs}$ if they induce the same partial order. The reason for this is that two equivalent clocks separate vertices in the same way into a sequence of sessions. We will sometimes abuse terminology and use "clock" to mean "clock equivalence class".

We next define a distortion function on clock equivalence classes. This will allow us to measure how far a given estimated clock is from the ground truth. Note that given an observed infection sequence $S_{obs}$, a clock cannot reverse the order of any pair of events, so that the standard Kendall $\tau$ distance between partial orders is not appropriate here.

**Definition 4** (Distortion function on clock pairs). *Consider two clocks $C_0, C_1$ with respect to an observed infection sequence $S_{obs}$. We define $\text{Dis}_{C_0, C_1}(i, j)$ to be the indicator that the clocks $C_0$ and $C_1$ order vertices*

*i and j differently (i.e., that the partial order on vertices induced by $C_b$ orders i and j and the partial order induced by $C_{1-b}$ does not, for b equal to either 0 or 1). If the clocks in question are clear from context, we may drop the subscript.*

*We define the following distortion measure on clock pairs:*

$$d_{S_{obs}}(C_0, C_1) = \frac{1}{\binom{n}{2}} \sum_{i<j} \text{Dis}_{C_0,C_1}(i,j). \tag{2}$$

**Main problem statement:** We finally come to the general problem that we would like to solve:

**Definition 5** (Clock estimation). *Fix a graph G, a small-scale cascade model $\mathcal{C}_0$, and a clock C. An infection sequence $S_{obs} \sim \mathcal{M}(C)$ is generated on G. Our goal is to produce an estimator $\hat{C} = \hat{C}(S_{obs})$ of C so as to minimize $\mathbb{E}[d_{S_{obs}}(C, \hat{C})]$ . This is called the* clock estimation problem.

The above definition implicitly assumes knowledge of the parameters of the small-scale cascade model. Estimation of these parameters has been studied extensively in the literature and need not necessarily come from cascade observations. We discuss this in the appendix, in Section E. Furthermore, knowledge of the initial conditions of the cascade is necessary in order to achieve an expected estimation error that tends to 0 in general. We thus assume that the initial network state is given to us. Under mild additional assumptions on the model (e.g., that $S_{sess}(0)$ consists of $\Theta(1)$ vertices chosen uniformly at random, and that the graph is sparse, so that $S_{sess}(0)$ is an independent set with high probability), the initial set $S_{sess}$ can be inferred with high probability. We discuss how to do this at length in the appendix, in Section F.

**Specific small-scale cascade model:** Having defined our multiscale generative model, we specify an example small-scale cascade model for our problem. Our approach generalizes beyond this one, as we will explain in Section 3.5.

We define the *discretized independent cascade (IC)* process.

**Definition 6** (Discretized independent cascade process). *We fix a graph G, an initial infection set $S_0$ of vertices in G (given by elements of $[n] = \{1, ..., n\}$), a number of timesteps K (giving the length of the session) and probability parameters $p_n$ and $p_e$, both in $[0,1]$. Here, $p_n$ denotes the probability of transmission of an infection across an edge, and $p_e$ denotes the probability of infection from an external source. Additionally, we fix a probability distribution D on $\mathbb{N}$ (for simplicity, we will choose the geometric distribution with mean 1).*

*When a node v becomes active, for every one of its susceptible neighbors w, it draws an independent Bernoulli($p_n$) random variable. If it is 1, v starts a timer by drawing a sample X from D, then defining $\tau_{v,w}$ to be the minimum of X and the number of timesteps remaining in the session. The vertex v then becomes inactive (moves to the I state). After $\tau_{v,w}$ timesteps, w becomes active.*

*In parallel, in a given timestep, each susceptible vertex becomes active with probability $p'_e = 1 - (1 - p_e)^{1/K}$, independent of anything else. This models infection by external circumstances, with probability $p_e$ over the entire session.*

*We denote by $S_j$ the set of nodes that become active in timestep j. The process terminates either after K steps or after all nodes are infected.*

There is a natural connection between the discretized IC model and the IC model defined in Kempe et al. (2003) via our two-scale model: when the small-scale model is the discretized IC model, the sequence $S_{sess} = (S_{sess}(0), S_{sess}(1), ..., S_{sess}(T))$ of sets of vertices infected in the sessions of our model is distributed according to the model in Kempe et al. (2003). In that model, there is a seed set of vertices $S_0$ that are active at time 0. In each timestep (our sessions), the vertices that were infected in the previous timestep become active, and the vertices that were active in the previous timestep become inactive but infected. Each active vertex chooses to infect each of its neighbors independently with probability $p_n$. Furthermore, each uninfected vertex becomes infected with probability $p_e$.

# 3 Main results: Algorithm, approximation and running time guarantees, generality

In this section, we present our proposed algorithm (Algorithm 1) for clock estimation, which we call *FastClock*. **We give full proofs of all theorems in Appendix B.**

FastClock takes as input a graph $G$, an observed infection sequence $S_{obs} = (S_{obs}(0), ..., S_{obs}(N))$ , and the parameters $\theta$ of the cascade model, including the initial infection set $S_{sess}(0)$ (see our discussion of this assumption in the previous section), but *not* including session lengths. The output of the algorithm is an estimated clock $\hat{C}$, which takes the form of a sequence of interval right endpoints $\hat{t}_0, \hat{t}_1, ..., \hat{t}_{\hat{N}} \in [[N]]$, for some $\hat{N}$ and is an estimate of the ground truth clock $C$ specified by $t_0, ..., t_N$.

Our algorithm proceeds by iteratively computing the estimate $\hat{t}_j$. In the $(j + 1)$-st iteration, to compute $\hat{t}_{j+1}$, it chooses the size of the next interval of the clock so as to match as closely as possible the expected number of newly infected nodes in the next session. We prove that the resulting clock estimate is very close, in terms of $d_{S_{obs}}(\cdot, \cdot)$ , to the ground truth clock, using concentration inequalities.

The correctness of FastClock is based on the following intuition: if we manage to correctly estimate $t_0, ..., t_j$, then we can estimate the conditional expected number of vertices infected in the $(j + 1)$-st session of the process (i.e., $|S_{sess}(j + 1)|$). We can show a conditional concentration result for $|S_{sess}(j + 1)|$ around its expectation. Thus, we output as our next clock interval endpoint $\hat{t}_{j+1}$ the smallest integer $t \geq \hat{t}_j$ for which the number of vertices in $\bigcup_{k=t_j+1}^{t} S_{obs}(t)$ does not exceed its conditional expectation, corrected by a small quantity. This quantity is determined by the concentration properties of the random variable $|S_{sess}(j + 1)|$ conditioned on the state of the process given by $S_{obs}(0), ..., S_{obs}(t_j)$. We choose the threshold to be such that, under this conditioning, the number of vertices infected in the next session is slightly less than it with probability tending exponentially to 1. Our approximation analysis illustrates that the approximation quality depends on the graph structure and the model parameters.

The significance of the approximation and running time results (Theorems 1 and 2 below for the independent cascade model) is that the clock parameter of our multiscale cascade model can be quickly estimated with provably high accuracy using relatively simple expected value calculations. The generality of our results (well beyond the IC model) is discussed in Section 3.5. As long as the expected number of nodes infected in the next session can be calculated efficiently, the FastClock algorithm can be adapted to a wide variety of cascade models.

## 3.1 The FastClock algorithm

Before we define our algorithm we introduce some necessary notation. For a session-level infection sequence $\tilde{S}$ and a session index $t \in |\tilde{S}|$, define $\sigma_t(\tilde{S})$ to be the $\sigma$-field generated by the event that the first $t$ session-level infection sets of the cascade process are given by $\tilde{S}_0, \tilde{S}_1, ..., \tilde{S}_t$. That is, the event in question is that $S_{sess}(0) = \tilde{S}_0, ..., S_{sess}(t) = \tilde{S}_t$. We also define $\mu_t(\tilde{S})$ to be $\mu_t(\tilde{S}) = \mathbb{E}[|S_{sess}(t + 1)| \mid \sigma_t(\tilde{S})]$. For a vertex $v$, we denote by $\mathcal{N}(v)$ the set of neighbors of $v$ in $G$. The algorithm is given in Algorithm 1.

After an initialization, the main loop in *FastClock* (Steps 5-11) iteratively determines the last infection event in the next session of the process (whose endpoint timestep we are trying to estimate), by estimating the expected number of nodes $\mu_t$ to be infected in that session (Step 6). The key step in this procedure is the computation of $\mu_t$, which we discuss next.

**Computing $\mu_t(\tilde{S})$ in the discretized IC model.** Let us be more precise in specifying how to compute $\mu_t(\tilde{S}) = \mathbb{E}[|S_{sess}(t + 1)| \mid \sigma_t(\tilde{S})]$ when the small-scale model is the discretized independent cascade model (see Definition 6). A node can be infected in one of two ways: through external factors (governed by $p_e$) or via transmission from a vertex in $\tilde{S}_t$ through an edge. In the latter case, the node must lie in the *frontier set* $\mathbb{F}_t(\tilde{S})$, defined as follows: $\mathbb{F}_t(\tilde{S})$ is the set $\mathbb{F}_t(\tilde{S}) = \mathcal{N}(\tilde{S}_t) \setminus \bigcup_{j=0}^{t} \tilde{S}_t$; i.e., it is the set of neighbors of $\tilde{S}_t$ that we believe to be uninfected at the beginning of cascade session $t$. See Example 5 in the appendix for an illustration of the definition of frontier sets.

For a set of vertices $W \subseteq [n]$ and a vertex $v \in [n]$, let $\deg_W(v)$ denote the number of edges incident on $v$ that are also incident on vertices in $W$. We can use linearity of expectation to derive a closed-form formula

---

**Algorithm 1:** FastClock

---

**Data:** Graph $G$, small-scale cascade model parameters $\theta$, observed infection sequence
$$S_{obs} = (S_{obs}(0), ..., S_{obs}(N))$$
**Result:** An estimated clock $\hat{C}$.
// An initially empty list for the estimated clock. This will eventually contain a
   sequence of *estimated endpoints* of clock intervals.

**1** Set $\hat{C} = ()$;
// $t$: index of the next estimated clock interval, i.e., $t$ is an index in $S_{sess}$, the
   session-level infection sequence.
// $t_{obs}$: the index in $S_{obs}$ of the beginning of the next estimated clock interval

**2** Set $t = 1, t_{obs} = \min\{j \leq N \ : \ | \cup_{k=0}^{j} S_{obs}(k)| = S_{sess}(0)\}$;
// $\tilde{S}$: the estimated infection sequence approximating the session-level sequence $S$.

**3** Set $\tilde{S}_0 = \cup_{k=0}^{t_{obs}} S_{obs}(k)$ ;

**4** Append $t_{obs}$ to $\hat{C}$;

**5 while** $t_{obs} \neq N$ **do**

    // Compute the expected number $\mu_t$ of infected nodes in a single session of the
       cascade process, starting from the state of the process estimated so far.

**6**     Set $\mu_t = \mathbb{E}[|S_{sess}(t+1)| \mid \sigma_t(\tilde{S}_0, \tilde{S}_1, ..., \tilde{S}_t)]$ ;

**7**     Set

$$t'_{obs} = t_{obs} + \max \left\{ \Delta \mid \sum_{i=t_{obs}+1}^{t_{obs}+\Delta} |S_{obs}(i)| \leq \mu_t \cdot (1 + \mu_t^{-1/3}) \right\} \tag{3}$$

**8**     Append $t'_{obs}$ to $\hat{C}$;

**9**     Set $\tilde{S}_t = \cup_{i=t_{obs}+1}^{t'_{obs}} S_{obs}(i)$;

**10**     Set $t = t + 1$;

**11**     Set $t_{obs} = t'_{obs}$;

**12 end**

**13 return** $\hat{C}$;

---

for $\mu_t(\tilde{S})$:

$$\mu_t(\tilde{S}) = p_e \cdot \left( n - |\mathbb{F}_t(\tilde{S})| - \sum_{j=0}^{t} |\tilde{S}_j| \right) + \sum_{v \in \mathbb{F}_t(\tilde{S})} (p_e + (1 - p_e)(1 - (1 - p_n)^{\deg_{\tilde{S}_t}(v)})). \tag{4}$$

A similar expression can be derived for the more general case where transmission probabilities across edges may differ from each other. The calculation of the summation $\sum_{j=0}^{t} |\tilde{S}_j|$ can be performed efficiently by keeping track of its value in the $t$-th iteration of the loop of the algorithm. In the $t$-th iteration, the value of the summation is updated by adding $|\tilde{S}_t|$ to the running total. Note also that this estimation will be the only difference in our algorithm when applied to alternative cascade models such as the linear threshold model.

### 3.2 Approximation guarantee for FastClock

Our first theorem gives an approximation guarantee for FastClock in the case where the small-scale model is the discretized IC model. It is subject to an assumption about the parameters of the graph model from which $G$ is sampled and about the parameters of the small-scale cascade model, which we state next. It is, however, important to note that FastClock itself does not assume anything about the graph. Furthermore, we generalize our guarantees beyond these assumptions in Section 3.5, Theorem 3.

**Assumption 1** (Assumptions on random graph model parameters). *We assume that $G \sim G(n, p)$ (i.e., that $G$ is sampled from the Erdős-Rényi model), where $p$ satisfies the following relation with the ground truth*

number of sessions $T + 1$: $p = o(n^{-\frac{T}{T+1}})$. and $p \geq C \log n / n$, for some $C > 1$. We substantially relax our modeling assumptions in Section 3.5.

The former condition on $p$ may be viewed as a constraint on $T$. It is natural in light of the fact that, together with our assumptions on $p_n$ and $p_e$ below, it implies that the cascade does not flood the graph, in the sense of infecting a $\Theta(1)$-fraction of nodes. Many cascades in practice do not flood the graph in this sense. The lower bounding condition on $p$ implies that the graph is connected with high probability. In terms of density, this covers graphs in the semi-sparse regime, wherein the number of edges grows superlinearly with the number of vertices. This is common in real networks with communities. We exhibit in Section H several real networks with average degrees that are well within the bounds of our assumption.

Regarding the small-scale cascade process, we assume that it is the discretized IC model from Definition 6, with the following constraints on the parameters: we assume that $p_n$ is some fixed positive constant and that $p_e = o(p)$. Our results also hold if $p_n$ is different for every edge $e$ (so that $p_n = p_n(e)$), provided that there are two positive constants $0 < c_0, c_1 < 1$ such that for every edge $e$, $p_n(e) \in [c_0, c_1]$.

The assumption that $p_n$ is constant with respect to $n$ is natural in the sense that, for many infectious processes, the probability of transmission from one node to another should not depend on the number of nodes. The assumption on $p_e$, the probability of infection from an external source, is reasonable when the cascade is overwhelmingly driven by network effects, rather than external sources.

**Theorem 1** (Main FastClock approximation theorem). *Suppose that Assumption 1 holds. We have, with probability at least $1 - e^{-\Omega(np)}$,*

$$d_{S_{obs}}(C, \hat{C}) = O((np)^{-1/3}), \tag{5}$$

*where we recall that $S_{obs}$ is the observed infection sequence generated by the two-scale cascade model parametrized by the ground truth clock $C$, $S_{sess}$ is the session-level infection sequence, and $\hat{C}$ is the estimate of $C$ that is output by FastClock.*

The proof of Theorem 1 uses an auxiliary result (Theorem 4 in the appendix, which we call the *FastClock utility theorem*) stating that with high probability, for every $i$, the intersection of the session-level infection sequence element $S_{sess}(i)$ with the estimated infection sequence element $\tilde{S}_i$ is asymptotically equivalent in cardinality to $S_{sess}(i)$ itself. We prove this by induction on the session index $i$, which requires a careful design of the inductive hypothesis. Given the utility theorem, the upper bound on the distortion $d_{S_{obs}}(C, \hat{C})$ follows by summing over all possible pairs $S_{sess}(i), S_{sess}(j)$ of infection sequence elements in $S_{sess}$, the session-level infection sequence, then summing over all vertex pairs $u \in S_{sess}(i), v \in S_{sess}(j)$. This inner sum is approximated using the utility theorem.

### 3.3 Running time analysis

We have a strong guarantee on the running time of FastClock in the independent cascade case. The running time of FastClock is asymptotically much smaller than that of the dynamic programming estimator from DiTursi et al. (2017).

**Theorem 2** (Running time of FastClock). *The FastClock algorithm for the case where the small-scale cascade model is the discretized independent cascade model runs in time $O(N + n + m)$, where $m$ is the number of edges in the input graph.*

### 3.4 Empirical results on synthetic graphs

In this section, we present empirical results on synthetic graphs and cascades. Our goal is to confirm the theoretical guarantees of *FastClock* and compare it to the *dynamic programming (DP)* algorithm optimizing a proxy of the maximum likelihood for observed cascades proposed by DiTursi et al. (2017). Our comparative analysis focuses on (i) *distance* of the estimated clock from the ground truth clock (see Definition 4) and (ii) empirical *running time* of both techniques. More extensive empirical results on synthetic and real graphs are included in the appendix, Section D.

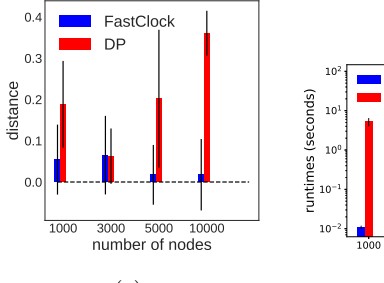
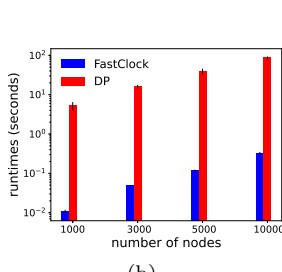
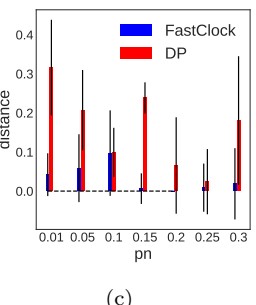
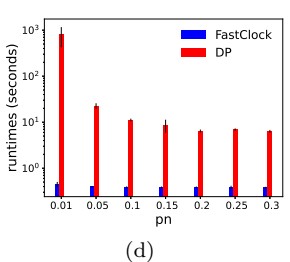

$\qquad$(a)$\qquad\qquad\qquad$(b)$\qquad\qquad\qquad$(c)$\qquad\qquad\qquad$(d)

Figure 2: Comparison of the distance and runtime of the estimated clocks by *FastClock* and the baseline DP from DiTursi et al. (2017) on Erdős–Rényi graphs (default parameters for all experiments: $p_n = 0.1$, $p_e = 10^{-7}$, $n = 3000$, $p = n^{-1/3}$, stretch $l = 2$ unless varying in the specific experiment). (a),(b): Varying graph size. (c),(d): Varying infection probability $p_n$.

We generate synthetic graphs using the Erdős–Rényi model (experiments using stochastic block models are included in Section D). We then generate synthetic cascades on each graph using the independent cascade (IC) model. Since both algorithms under consideration are invariant to infection timers in the small-scale model, we determine the set of vertices infected in each session according to IC model of Kempe et al. (2003), then assign to each infected vertex a uniformly random infection time within its session, which we fix to have length $l$ (some integer which we call the *stretch factor* of our cascade). As in our theorems, we denote by $S_{sess}$ the session-level infection sequence and by $S_{obs}$ the observed process infection sequence. These implicitly specify a ground-truth clock $C$. We note that while all of our experiments involve sessions with uniform length, our theoretical contributions are more general. We then employ both *FastClock* and the maximum likelihood proxy algorithm to estimate the ground truth clock from $S_{obs}$. We draw 50 samples for each setting and report average and standard deviation for both running time and quality of estimations for each setting.

**Experiments on Erdős–Rényi graphs.** We report a subset of the results of our experiment on Erdős-Rényi graphs in Figure 2. With increasing graph size *FastClock*'s distance from the ground truth clock diminishes (as expected based on Theorem 1), while that of DP increases (Fig. 2(a)). Note that DP optimizes a proxy to the cascade likelihood and in our experiments tend to associate too many early timesteps with early sessions, which for large graph sizes results in incorrect recovery of the ground truth clock. Similarly, *FastClock*'s estimate quality is better than that of DP for varying on $p_n$ (Fig. 2(c)), graph density (Fig. 4(e) in the appendix) and stretch factor for the cascades (Fig. 4(g) in the appendix), with distance from ground truth close to 0 for regimes aligned with the key assumptions we make for our main results (Assumption 1 or, more generally, 2). In addition to superior accuracy, *FastClock*'s running time scales linearly with the graph size and is orders of magnitude smaller than that of DP for sufficiently large instances (Figs. 2(b), 2(d)).

### 3.5 Generality of FastClock

We have shown that FastClock achieves small expected distortion on the clock recovery problem when the small-scale model is as in Definition 6. However, the algorithm works substantially more generally – all that is needed is concentration of the number of vertices infected in each session, given previous session, around its conditional expectation, along with concentration of the frontier size in each session. In particular, the model-dependent parts of the proof of Theorem 1 lie entirely in the proof of Theorem 4 and the associated auxiliary lemmas.

Below, we formulate sufficient conditions on our model to guarantee these properties. Our discussion culminates in Assumption 2, which is used as the main hypothesis in Theorem 3.

**Concentration of the number of vertices infected in each session** A sufficient condition for the required concentration property to hold is a *martingale difference property*: for any session index $t$, we may write $|S_{sess}(t)|$ as a sum of indicators of vertex infection events: letting $X(v, t)$ denote the indicator

that $v \in S_{sess}(t)$, we have $|S_{sess}(t)| = \sum_v X(v, t)$. We define $Z(S_{sess}, t, v)$ as follows: we first order the vertices in $\mathbb{F}(S_{sess}, t)$ arbitrarily and denote them by $v_1, ..., v_{|\mathbb{F}(S_{sess}, t)|}$. Then we define $Z(S_{sess}, t, j) = \mathbb{E}[|S_{sess}(t)| \mid \sigma_{t-1}(S_{sess}), X(v_1, t), ..., X(v_j, t)]$.

Then $\{Z(S_{sess}, t, j)\}_{j=1}^{|\mathbb{F}(S_{sess}, t)|}$ is the Doob martingale with respect to the natural filtration on the vertex infection events in the frontier up to vertex $v_j$. By the Azuma-Hoeffding inequality, the desired concentration property holds whenever

$$|Z(S_{sess}, t, j) - Z(S_{sess}, t, j-1)| = O(|\mathbb{F}(S, t)|^{1/2 - const}), \tag{6}$$

for all $t, j$, with probability 1, where the $O(\cdot)$ is uniform in all parameters and $0 < const < 1/2$. Intuitively, this means that knowledge of whether or not a given susceptible vertex becomes infected in session $t$ does not substantially alter our best guess of the total number of vertices infected in that session. This encompasses our IC-based model but is substantially more general, since it allows for, e.g., weak dependence among infection events.

**Concentration of the frontier size**  Concentration of the frontier size at each step is assured when $G$ is drawn from any random graph model coming from a *sparse graphon* (Borgs et al., 2017) $W$ with density parameter $\rho_n$ within the range of $p$ specified by Assumption 1 and with all entries bounded away from 0. More precisely, all that is needed is concentration of the frontier size, conditioned on the latent positions of all nodes. This concentration follows immediately from a Chernoff bound.

To make this rigorous, we first explain the sparse graphon framework. This is encapsulated in Definitions 7 and 8 below.

**Definition 7** (Graphon). *A graphon is a symmetric, Lebesgue-measurable function $W : [0, 1]^2 \to [0, 1]$.*

**Definition 8** (Sparse random graph model associated with a graphon). *The sparse random graph model $G(W, \rho_n)$ with sparsity parameter $\rho_n$ (a function of $n$ whose codomain is $[0, 1]$) associated with the graphon $W$ is the following distribution on graphs with $n$ vertices $\{1, 2, ..., n\}$: first, $n$ numbers $X_1, ..., X_n$ are sampled uniformly at random from $[0, 1]$. Conditioned on these, an edge exists between vertices $i, j$ independently of anything else with probability $\rho_n \cdot W(X_i, X_j)$.*

We can now formulate the more general assumptions under which we can prove an accuracy guarantee for FastClock.

**Assumption 2** (General assumptions on the random graph model and the small-scale model). *We assume that the graph $G$ is sampled from a sparse graphon $G(W, \rho_n)$, where $\rho_n$ satisfies the same properties that $p$ does in Assumption 1. Furthermore, we assume that there exist two constants $p_0, p_1 \in (0, 1)$ such that $p_0 \leq W(x, y) \leq p_1$ for all $x, y \in [0, 1]$. We denote $p = p_0 \cdot \rho_n$.*

*We assume that the small-scale cascade model $C_0$ satisfies the following conditions:*

   *a) The martingale difference condition described in (6).*

   *b) At any timestep $t$ and for any vertex $v$ in the current frontier, the conditional probability that $v$ becomes infected during timestep $t$ depends only on the set of its active neighbors.*

   *c) There exists a universal constant $c_0 > 0$ such that in any timestep, the probability of infection of any frontier vertex is bounded below by $c_0$.*

Then our generalized accuracy guarantee for FastClock is given in the following theorem.

**Theorem 3** (FastClock accuracy guarantee, generalized). *Suppose that Assumption 2 holds. We have, with probability at least $1 - e^{-\Omega(n\rho_n)}$,*

$$d_{S_{obs}}(C, \hat{C}) = O((n\rho_n)^{-1/3}). \tag{7}$$

We note that Theorem 1 is a consequence of Theorem 3. To show that this is the case, it suffices to check that Assumption 2 follows from Assumption 1. Since $G$ is sampled from an Erdős-Rényi model with edge

probability $p = p_0 \rho_n$, it is equivalently a sample from a sparse graphon model $W$ satisfying $W(x, y) = p_0$ for all $x, y \in [0, 1]$, and with $\rho_n$ as a sparsity parameter. Regarding the assumptions about the small-scale model, assumptions (b) and (c) hold for the IC model trivially: specifically, the constant $c_0$ is simply the minimum transmission probability for any edge. Assumption (a), the martingale difference condition, is also checked simply: in the IC model, the terms of the sum $\sum_j X(v, t)$ defining $|S_{sess}(t)|$ are independent, and so it may be checked that $|Z(S_{sess}, t, j) - Z(S_{sess}, t, j-1)| \leq |X(v_j, t) - \mathbb{E}[X(v_j, t) \mid \sigma_{t-1}(S_{sess})]| \leq 1$, where the last step is because $X(v_j, t)$ is an indicator function. This martingale difference bound is *substantially less than the required one*. This completes the proof that Assumption 2 follows from Assumption 1.

## 4   Conclusions and future work

We have formulated a generative model and statistical estimation framework for network spreading processes whose activity periods (which we called *sessions*) are intermittent. The model is parametrized by a *clock*, which encodes the large-scale behavior of the process. We showed that this clock can be estimated with high accuracy and low computational cost, subject to certain natural constraints on the structure of the underlying graph and on the small-scale cascade model: in essence, these must be such that the graph is an expander with appropriate parameters; that, conditioned on an estimated current state of the process at any time, the expected number of vertices infected in the next session is immune to small errors in the estimated state; and that the number of vertices infected in the next session is well-concentrated around its conditional expected value. We empirically showed that the FastClock algorithm is superior in accuracy and running time to the current state of the art dynamic programming algorithm. Furthermore, unlike this baseline, FastClock comes with theoretical accuracy guarantees. Our results hold for a broad class of small-scale cascade models, provided that they satisfy a certain martingale difference property. Furthermore, the class of random graph models for which our guarantees hold is similarly broad.

We intend to pursue further work on this problem: most pressingly, our empirical results and intuition derived from our theorems indicate that FastClock may not perform accurately when the graph contains very sparse cuts (so that it is not an expander graph, and it cannot have been generated by a sparse graphon model with the assumed parameter ranges, except with very small probability). Our empirical evidence is consistent with the conjecture that the degradation of performance on real networks in comparison to the DP algorithm is a result of sparse cuts. In such graphs, we envision that FastClock can be used within communities as a subroutine of a more complicated algorithm. Further work is needed to determine whether accuracy and computational efficiency can be achieved for such graphs. Additionally, our model can be extended to the case where different subsets of nodes behave according to distinct clocks. It is important to clarify the information-theoretic limits of clock estimation and related hypothesis testing questions in this scenario.

## 5   Acknowledgements

This research is funded by an academic grant from the National Geospatial-Intelligence Agency (Award No. # HM0476-20-1-0011, Project Title: Optimizing the Temporal Resolution in Dynamic Graph Mining). Approved for public release, NGA-U-2022-02239. The work was also supported in part by the NSF Smart and Connected Communities (SC&C) CMMI grant 1831547 and by NSF CCF grant number #2212327.

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

## A  Glossary of notation

Here we collect the notation that is used in the main body of the paper and in the proofs in the appendix.

1. $\mathcal{N}(S)$: Neighborhood of the set $S$ of vertices in a given graph.

2. $S_{sess} = (S_{sess}(0), S_{sess}(1), ..., S_{sess}(T))$ – A session-level infection sequence with $T + 1$ sessions. Each $S_{sess}(j)$ is a subset of vertices, and $S_{sess}(i) \cap S_{sess}(j) = \emptyset$ for $i \neq j$. We denote by $|S_{sess}|$ the number of sessions of $S_{sess}$: $T + 1$.

3. $S_{obs} = (S_{obs}(0), S_{obs}(1), ..., S_{obs}(N))$ – The observed infection sequence, as generated by our multi-scale cascade model.

4. $C$ – The ground-truth clock in our estimation problem.

5. $\hat{C}$ – The clock estimated by our algorithm.

6. $\tilde{S}$ – The estimate of the session-level infection sequence $S_{sess}$ induced by our estimate $\hat{C}$ of the clock $C$ applied to the observed infection sequence $S_{obs}$.

7. $\sigma_t(S)$, for an infection sequence $S$ and a timestep index $t \in |S|$ – The $\sigma$-field generated by the event that the first $t$ session-level infection sets of the process are given by $S_0, ..., S_t$.

8. $\mu_t(S)$, for an infection sequence $S$ and a timestep index $t \in |S|$ – $\mathbb{E}[|S_{sess}(t+1)| \mid \sigma_t(S)]$. This is the expected number of vertices infected in the $t+1$st session, given the session-level infection sequence up to and including timestep $t$.

9. $N$ – The index of the last observed infection set. That is, $|S_{obs}| = N + 1$.

10. $T$ – The index of the last session-level infection set. That is, $|S_{sess}| = T + 1$.

11. $n$ – The size of the graph.

12. $p_n$ – The probability in the IC model of transmission across an edge in a single timestep.

13. $p_e$ – The probability of infection of a vertex in a single timestep by a non-network source.

14. $R(S, i)$ – For an infection sequence $S$ and an index $i$, define the $i$th *running sum* to be

$$R(S, i) = \bigcup_{j \leq i} S_j. \tag{8}$$

15. $\mathbb{F}_i(S) = \mathbb{F}(S, i)$ – For an infection sequence $S$ and an index $i \in \{0, 1, ..., |S|\}$, define the $i$th frontier set to be

$$\mathbb{F}(S, i) = \mathcal{N}(S_i) \setminus R(S, i). \tag{9}$$

The $i$th frontier with respect to $S$ is the set of neighbors of vertices infected in session $i$ that have not infected by the end of session $i$.

16. $\mathcal{CF}(S, i)$ – The candidate frontier set at the end of session $i$ in infection sequence $S$. That is, this is

$$\mathcal{CF}(S, i) = [n] \setminus R(S, i). \tag{10}$$

Note that $\mathbb{F}(S, i) \subseteq \mathcal{CF}(S, i)$.

17. $\mathcal{CCF}(S, \tilde{S}, i, j)$ – The common candidate frontier:

$$\mathcal{CCF}(S, \tilde{S}, i, j) = \mathcal{CF}(S, i) \cap \mathcal{CF}(\tilde{S}, j). \tag{11}$$

## B    Proofs

In this section, we give full proofs of all results.

### B.1    Proof of Theorem 1

To prove the main FastClock approximation theorem, we start by characterizing the growth of $\mu_i(S_{sess})$ and $|S_{sess}(i)|$ as a function of $n$ and $i$. Note that this is a result about the independent cascade process, not the FastClock algorithm.

**Lemma 1** (Growth of $\mu_i(S_{sess})$ and $|S_{sess}(i)|$). *We have that, with probability at least $1 - e^{-np}$, for all $i \leq T$,*

$$\mu_i(S_{sess}) = \Theta((np)^{i+1}), \tag{12}$$

*where the $\Theta(\cdot)$ is uniform in $i$. Furthermore, with probability at least $1 - e^{-np}$, we have*

$$|S_{sess}(i)| = \Theta((np)^i) \tag{13}$$

*for every $i$.*

*Proof.* We prove this by induction on $i$ and use the formula (4) throughout.

**Base case ($i = 0$):**  In the base case, we are to verify that $\mu_0(S_{sess}) = \Theta(np)$. The first term of (4) is non-negative and at most $p_e \cdot n$. By our assumption, we have that $p_e = o(p_n)$, implying that the first term is $o(np)$. Thus, it remains for us to show that the second sum is $\Theta(np)$. The dominant contribution comes from the second term of each term of the sum:

$$\sum_{v \in \mathbb{F}_0(S_{sess})} (p_e + (1 - p_e)(1 - (1 - p_n))^{\deg_{S_{sess}(0)}(v)}) = \Theta\left( \sum_{v \in \mathbb{F}_0(S_{sess})} 1 - (1 - p_n)^{\deg_{S_{sess}(0)}(v)} \right) \tag{14}$$

$$= \Theta\left( |\mathbb{F}_0(S_{sess})| - \sum_{v \in \mathbb{F}_0(S_{sess})} (1 - p_n)^{\deg_{S_{sess}(0)}(v)} \right). \tag{15}$$

In the final expression above, the remaining sum is lower bounded by 0 and upper bounded by $C|\mathbb{F}_0(S_{sess})|$ for some constant $C < 1$, since each term is between 0 and $C$. Thus, we have shown that, with probability exactly 1,

$$\mu_0(S_{sess}) = \Theta(|\mathbb{F}_0(S_{sess})|) + o(np). \tag{16}$$

Since $|\mathbb{F}_0(S_{sess})|$ is the set of uninfected neighbors of all vertices in $S_{sess}(0)$, and, by assumption, $|S_{sess}(0)| = \Theta(1)$, we have that with probability at least $1 - e^{-np}$,

$$|\mathbb{F}_0(S_{sess})| = \Theta(np). \tag{17}$$

Thus, we have

$$\mu_0(S_{sess}) = \Theta(np) \tag{18}$$

with probability $\geq 1 - e^{-np}$. Conditioning on this event (which is only an event dealing with the graph structure), we have that $|S_{sess}(1)| \sim \text{Binomial}(\Theta(np), p_n)$, and a Chernoff bound gives us that with probability $1 - e^{-\Omega(np)}$, $|S_{sess}(1)| = \Theta(np)$, as desired. This completes the proof of the base case.

**Induction ($i > 0$, and we verify the inductive hypothesis for $i$):** We assume that $\mu_j(S_{sess}) = \Theta((np)^{j+1})$ and $|S_{sess}(j+1)| = \Theta((np)^{j+1})$ for $j = 0, 1, ..., i-1$. We must verify that it holds for $j = i$, with probability at least $1 - e^{-np}$. As in the base case, the first term of (4) is $O(np_e) \ll np \ll (np)^{i+1}$. It is, therefore, negligible with probability 1. The second term again provides the dominant contribution and is easily seen to be $\Theta(|\mathbb{F}_i(S_{sess})|)$, just as in the base case. Thus, it remains to show that $|\mathbb{F}_i(S_{sess})| = \Theta((np)^{i+1})$ with probability at least $1 - e^{-\Omega(np)}$, which implies the desired result for $\mu_i(S_{sess})$. The inductive hypothesis implies that $|S_{sess}(i)| = \Theta((np)^i)$, and the number of uninfected vertices is $n - \sum_{j=0}^{i} |S_{sess}(j)| = n - \Theta((np)^{i+1})$. Since $i \leq T - 1$, this is asymptotically equivalent to $n$.

Now, conditioned on the first $i$ elements of $S_{sess}$, the $i$th frontier $|\mathbb{F}_i(S_{sess})| \sim \text{Binomial}(n \cdot (1 - o(1)), 1 - (1-p)^{|S_{sess}(i)|})$. Thus, with probability at least $1 - e^{-\Omega((np)^i)}$, we have

$$|\mathbb{F}_i(S_{sess})| = \Theta(n \cdot (1 - (1-p)^{|S_{sess}(i)|})). \tag{19}$$

Now,

$$1 - (1-p)^{|S_{sess}(i)|} = 1 - (1-p)^{(np)^i}. \tag{20}$$

Since $p = o(1)$, we have

$$1 - (1-p)^{(np)^i} \sim 1 - e^{-p^{i+1}n^i} \tag{21}$$

Now, using the fact that the fact that $p = o(n^{-\frac{T}{T+1}})$, we have

$$p^{i+1}n^i = o(n^{-\frac{T}{T+1}(i+1)+i}), \tag{22}$$

from our assumption on the growth of $p$. Thus, in particular,

$$p^{i+1}n^i = o(1). \tag{23}$$

This implies that

$$1 - e^{-p^{i+1}n^i} = 1 - (1 - p^{i+1}n^i)(1 + O(p^{i+1}n^i)) = p^{i+1}n^i(1 + o(1)). \tag{24}$$

Thus, with probability at least $1 - e^{-\Omega((np)^i)}$,

$$|\mathbb{F}_i(S_{sess})| = \Theta((np)^{i+1}), \tag{25}$$

which implies that

$$\mu_i(S_{sess}) = \Theta((np)^{i+1}). \tag{26}$$

By concentration of $|S_{sess}(i)|$, we then have that with probability at least $1 - e^{-\Omega(\mu_i(S_{sess}))}$,

$$|S_{sess}(i)| = \Theta((np)^{i+1}), \tag{27}$$

as desired.

**Completing the proof** Let $G_i$ be the event that the inductive hypothesis holds for index $i = 0, 1, ..., T-1$. Then we have

$$\Pr[\cap_{i \geq 0} G_i] = \Pr[G_0] \cdot \prod_{i \geq 1} \Pr[G_i \mid \cap_{j=0}^{i-1} G_j] \geq \prod_{i=0}^{T-1} (1 - e^{-\Omega((np)^i)}) 1 - e^{-\Omega((np))}. \tag{28}$$

This completes the proof. $\qquad\square$

Next, we state and prove a utility theorem (Theorem 4 below). To state it, we need some notation: our estimated clock $\hat{C}$ induces an estimate $\tilde{S}$ of the ground truth session-level infection sequence $S_{sess}$. In particular, $\tilde{S}$ is the unique infection sequence such that distorting $\tilde{S}$ according to $\hat{C}$ yields $S_{obs}$ as an observed infection sequence.

**Theorem 4** (Main FastClock analysis utility theorem). *We have that with probability $1 - e^{-\Omega(np)}$, for every $i \leq T-1$,*

$$|S_{sess}(i) \cap \tilde{S}_i| = |S_{sess}(i)| \cdot (1 - O((np)^{-1/3})). \tag{29}$$

We will prove this theorem by induction on $i$. The inductive hypothesis needed is subtle, as a na ive hypothesis is too weak. To formulate it and to prove our result, we need some notation: for an infection sequence $W$, we define the $i$th running sum to be

$$R(W, i) = \bigcup_{j=0}^{i} W_j. \tag{30}$$

We define the *frontier* and *running sum discrepancy sets* between two session-level infection sequences $S, \hat{S}$ as follows:

$$\Delta\mathbb{F}(S, \hat{S}, i, j) = \mathbb{F}_i(S) \triangle \mathbb{F}_j(\hat{S}) \tag{31}$$

$$\Delta R(S, \hat{S}, i, j) = R(S, i) \triangle R(\hat{S}, j), \tag{32}$$

where $\triangle$ denotes the symmetric difference between two sets.

We define the *candidate frontier* at index $i$ in infection sequence $S$ to be

$$\mathcal{CF}(S, i) = [n] \setminus R(S, i). \tag{33}$$

This is the set of vertices that are not yet infected after index $i$.

We define the *common candidate frontier* to be

$$\mathcal{CCF}(S, \hat{S}, i, j) = \mathcal{CF}(S, i) \cap \mathcal{CF}(\hat{S}, j). \tag{34}$$

With this notation in hand, we define the following inductive hypotheses:

**Hypothesis 1.** *There is a small discrepancy between the running sums of the true and estimated clocks:*

$$||R(S_{sess}, i)| - |R(\tilde{S}, i)|| \leq f_1(n, i), \tag{35}$$

*where we set, with foresight, $f_1(n, i) = \mu_{i-1}(S_{sess})^{.66} = o(\mu_{i-1}(S_{sess})^{2/3})$.*

**Hypothesis 2.** *There is a small discrepancy between $S_{sess}(i)$ and $\tilde{S}_i$:*

$$1 - \frac{|S_{sess}(i) \cap \tilde{S}_i|}{|S_{sess}(i)|} \leq f_2(n, i), \tag{36}$$

*where we set, with foresight, $f_2(n, i) = D \cdot \mu_{i-1}(S_{sess})^{-1/3}$, for some large enough constant $D$.*

We will use these to prove Theorem 4. The base case and inductive steps are proven in Propositions 1 and 2 below. First, we start by proving an upper bound (Theorem 5) on the following difference:

$$|\mu_i(S_{sess}) - \mu_i(\tilde{S})|. \tag{37}$$

In essence, the upper bound says that at any given clock time step, the expected number of nodes infected in the next timestep is almost the same according to both the true and estimated clock. This will later be used to verify the two inductive hypotheses stated above.

**Theorem 5** (Upper bound on (37))**.** *Granted the inductive hypotheses explained above, we have that*

$$|\mu_i(S_{sess}) - \mu_i(\tilde{S})| \leq p\mu_{i-1}(S_{sess})^{2/3}\mu_i(S_{sess}), \tag{38}$$

*with probability* $\geq 1 - e^{-\Omega(\mu_i(S_{sess}))}$.

*Proof.* To upper bound (37), we apply the triangle inequality to (4) to get

$$|\mu_i(S_{sess}) - \mu_i(\tilde{S})| \leq p_e \cdot \left||\mathbb{F}_i(S_{sess})| - |\mathbb{F}_i(\tilde{S})|\right| \tag{39}$$

$$+ p_e \left|\sum_{j=0}^{i} |\tilde{S}_j| - \sum_{j=0}^{i} |S_{sess}(j)|\right| \tag{40}$$

$$+ \left|\sum_{v \in \mathbb{F}_i(S_{sess})} Q(i, S_{sess}, v) - \sum_{v \in \mathbb{F}_i(\tilde{S})} Q(i, \tilde{S}, v)\right|, \tag{41}$$

where $Q(i, S_{sess}, v) = p_e + (1 - p_e)(1 - (1 - p_n)^{\deg_{S_{sess}(i)}(v)})$.

We will upper bound each of the three terms (39), (40), and (41) separately.

**Upper bounding (39) by** $O(p_e|\mathbb{F}_i(S_{sess})|)$**:** We first note that

$$\left||\mathbb{F}_i(S_{sess})| - |\mathbb{F}_i(\tilde{S})|\right| \leq |\Delta\mathbb{F}(S_{sess}, \tilde{S}, i, i)|. \tag{42}$$

So it is enough to upper bound the frontier discrepancy set cardinality. In order to do this, we decompose it as follows:

$$|\Delta\mathbb{F}(S_{sess}, \tilde{S}, i, i)| = |\Delta\mathbb{F}(S_{sess}, \tilde{S}, i, i) \cap \Delta R(S_{sess}, \tilde{S}, i, i)| + |\Delta\mathbb{F}(S_{sess}, \tilde{S}, i, i) \cap \mathcal{CCF}(S_{sess}, \tilde{S}, i, i)|. \tag{43}$$

This decomposition holds for the following reason: let $v$ be a vertex in the frontier discrepancy set $\Delta\mathbb{F}(S_{sess}, \tilde{S}, i, i)$. Suppose, further, that $v$ is not in the common candidate frontier for $S_{sess}(i), \tilde{S}_i$ (so it does not contribute to the second term on the right-hand side of (43)). We will show that it must be a member of $\Delta R(S_{sess}, \tilde{S}, i, i)$, which will complete the proof of the claimed decomposition. Then $v$ must be a member of at least one of $R(S_{sess}, i), R(\tilde{S}, i)$ (i.e., it must already be infected in at least one of these). If it were a member of both, then it would not be a member of either frontier, so it could not be a member of the frontier discrepancy set. Thus, $v$ is only a member of one of $R(S_{sess}, i)$ or $R(\tilde{S}, i)$. This implies that $v \in \Delta R(S_{sess}, \tilde{S}, i, i)$. This directly implies the claimed decomposition (43).

We now compute the expected value of each term of the right-hand side of (43), where the expectation is taken with respect to the graph $G$. After upper bounding the expectations, standard concentration inequalities will complete our claimed bound on the size of the frontier discrepancy set.

In the first term, the size of the intersection of the frontier discrepancy with the running sum discrepancy is simply the number of vertices in the running sum discrepancy set that have at least one edge to some vertex in $S_{sess}(i)$ (here we assume, without loss of generality, that $|R(S_{sess}, i)| \leq |R(\tilde{S}, i)|$). Using linearity of expectation, the expected number of such vertices is

$$\mathbb{E}[|\Delta\mathbb{F}(S_{sess}, \tilde{S}, i, i) \cap \Delta R(S_{sess}, \tilde{S}, i, i)|] = |\Delta R(S_{sess}, \tilde{S}, i, i)| \cdot (1 - (1 - p)^{|S_{sess}(i)|}). \tag{44}$$

Here $(1 - (1-p)^{|S_{sess}(i)|})$ is the probability that, for a fixed vertex $w \in \Delta R(S_{sess}, \tilde{S}, i, i)$, there is at least one edge between $w$ and some vertex in $S_{sess}(i)$.

We compute the expected value of the second term of (43) as follows.

We claim that

$$\Delta \mathbb{F}(S_{sess}, \tilde{S}, i, i) \cap \mathcal{CCF}(S_{sess}, \tilde{S}, i, i) \subseteq \mathcal{CCF}(S_{sess}, \tilde{S}, i, i) \cap (\mathcal{N}(\Delta R(S_{sess}, \tilde{S}, i, i)) \setminus \mathcal{N}(S_{sess}(i))). \quad (45)$$

To show this, let $v \in \Delta \mathbb{F}(S_{sess}, \tilde{S}, i, i) \cap \mathcal{CCF}(S_{sess}, \tilde{S}, i, i)$. The fact that $v$ is in the frontier discrepancy set means that it has an edge to exactly one of $S_{sess}(i), \tilde{S}_i$. This implies that it has an edge to the running sum discrepancy set. Recalling that we assumed wlog that $|R(S_{sess}, i)| \leq |R(\tilde{S}, i)|$, we must have that $\Delta R(S_{sess}, \tilde{S}, i, i) \cap S_{sess}(i) = \emptyset$, and so we must also have that there are no edges from $v$ to $S_{sess}(i)$. This completes the proof of the claimed set inclusion. This implies that

$$\mathbb{E}[|\Delta \mathbb{F}(S_{sess}, \tilde{S}, i, i) \cap \mathcal{CCF}(S_{sess}, \tilde{S}, i, i)|] \leq \mathbb{E}[|\mathcal{CCF}(S_{sess}, \tilde{S}, i, i) \cap (\mathcal{N}(\Delta R(S_{sess}, \tilde{S}, i, i)) \setminus \mathcal{N}(S_{sess}(i)))|]. \quad (46)$$

As above, the expectation is taken with respect to the random graph $G$.

For a single vertex in the common candidate frontier, the probability that it lies in the frontier discrepancy set is thus at most

$$(1 - (1-p)^{|\Delta R(S_{sess}, \tilde{S}, i, i)|}) \cdot (1-p)^{|S_{sess}(i)|}. \quad (47)$$

Thus, using linearity of expectation, the expected size of the second term in (43) is upper bounded by

$$\mathbb{E}[|\Delta \mathbb{F}(S_{sess}, \tilde{S}, i, i) \cap \mathcal{CCF}(S_{sess}, \tilde{S}, i, i)| \mid \sigma_i(S_{sess})] \quad (48)$$

$$\leq |\mathcal{CCF}(S_{sess}, \tilde{S}, i, i)| \cdot (1 - (1-p)^{|\Delta R(S_{sess}, \tilde{S}, i, i)|}) \cdot (1-p)^{|S_{sess}(i)|}. \quad (49)$$

Combining (44) and (49) and defining $q = 1 - p$, we have the following expression for the expected size of the frontier discrepancy set:

$$\mathbb{E}[|\Delta \mathbb{F}(S_{sess}, \tilde{S}, i, i)|] \quad (50)$$

$$= |\Delta R(S_{sess}, \tilde{S}, i, i)| \cdot (1 - q^{|S_{sess}(i)|}) \quad (51)$$

$$+ |\mathcal{CCF}(S_{sess}, \tilde{S}, i, i)| \cdot (1 - q^{|\Delta R(S_{sess}, \tilde{S}, i, i)|}) \cdot q^{|S_{sess}(i)|}. \quad (52)$$

We would like this to be $O(\mathbb{E}[|\mathbb{F}_i(S_{sess})| \mid \sigma_i(S_{sess})])$. Note that $\mathbb{E}[|\mathbb{F}_i(S_{sess})| \mid \sigma_i(S_{sess})]$ can be expressed as follows:

$$\mathbb{E}[|\mathbb{F}_i(S_{sess})| \mid \sigma_i(S_{sess})] = (|\Delta R(S_{sess}, \tilde{S}, i, i)| \quad (53)$$

$$+ |\mathcal{CCF}(S_{sess}, \tilde{S}, i, i)|) \cdot (1 - q^{|S_{sess}(i)|}). \quad (54)$$

The intuition behind (50) being $O(\mathbb{E}[|\mathbb{F}_i(S_{sess})| \mid \sigma_i(S_{sess})])$ is as follows: the $\Delta R$ term is exactly the same as in (53). However, this term is negligible compared to the common candidate frontier term in both expected values. The second term, (52), can be asymptotically simplified as follows: we have

$$1 - q^{|\Delta R(S_{sess}, \tilde{S}, i, i)|} = 1 - (1-p)^{|\Delta R(S_{sess}, \tilde{S}, i, i)|} \quad (55)$$

$$\sim 1 - (1-p) \cdot |\Delta R(S_{sess}, \tilde{S}, i, i)|) \quad (56)$$

$$= p \cdot |\Delta R(S_{sess}, \tilde{S}, i, i)| \quad (57)$$

$$= p|S_{sess}(i)| \cdot \frac{|\Delta R(S_{sess}, \tilde{S}, i, i)|}{|S_{sess}(i)|} \quad (58)$$

$$\sim (1 - q^{|S_{sess}(i)|}) \cdot \frac{|\Delta R(S_{sess}, \tilde{S}, i, i)|}{|S_{sess}(i)|}. \quad (59)$$

Here, we have used the following facts:

- For the first asymptotic equivalence, we used the fact that $p|\Delta R(S_{sess}, \tilde{S}, i, i)| = o(1)$. More precisely, we have from the inductive hypothesis that

$$|\Delta R(S_{sess}, \tilde{S}, i, i)| = o(\mu_{i-1}(S_{sess})^{0.66}) = o((np)^{i \cdot 0.66}), \tag{60}$$

so we have

$$p|\Delta R(S_{sess}, \tilde{S}, i, i)| = o(p^{0.66i+1} n^{0.66i}) = o(n^{-T/(T+1)+0.66i/(T+1)}), \tag{61}$$

which is polynomially decaying in $n$.

- For the second asymptotic equivalence, we used the fact that $p|S_{sess}(i)| = o(1)$. More precisely, this comes from the fact that

$$p|S_{sess}(i)| = O(p(np)^i) = O(p^{i+1} n^i). \tag{62}$$

Now, we use the fact that $p = o(n^{-\frac{T}{T+1}})$:

$$p^{i+1} n^i = o(n^{-\frac{T}{T+1}(i+1)+i}), \tag{63}$$

from our assumption on the growth of $p$. Now, we need to show that the exponent is sufficiently negative and bounded away from 0.

$$-\frac{T}{T+1}(i+1) + i = \frac{-T \cdot (i+1) + i \cdot (T+1)}{T+1} = \frac{-T+i}{T+1} \leq \frac{-1}{T+1}. \tag{64}$$

We have used the fact that $i \leq T - 1$. Now, the constraints that we imposed on $p$ imply that $T = o(\log n)$, so

$$n^{\frac{-1}{T+1}} = e^{\frac{-\log n}{T+1}} = o(1), \tag{65}$$

as desired, since the exponent tends to $-\infty$ as $n \to \infty$.

Let us be more precise about what we proved so far. We have

$$\mathbb{E}[|\Delta\mathbb{F}(S_{sess}, \tilde{S}, i, i)| \mid \sigma_i(S_{sess})] \sim (1 - q^{|S_{sess}(i)|}) \cdot |\mathcal{CCF}(S_{sess}, \tilde{S}, i, i)| \cdot \left(\frac{|\Delta R|}{|\mathcal{CCF}|} + \frac{|\Delta R|}{|S_{sess}(i)|} q^{|S_{sess}(i)|}\right). \tag{66}$$

Meanwhile,

$$\mathbb{E}[|\mathbb{F}_i(S_{sess})| \mid \sigma_i(S_{sess})] = (1 - q^{|S_{sess}(i)|}) \cdot |\mathcal{CCF}| \cdot \left(1 + \frac{|\Delta R|}{|\mathcal{CCF}|}\right). \tag{67}$$

We have that

$$\frac{\mathbb{E}[|\Delta\mathbb{F}_i| \mid \sigma_i(S_{sess})]}{\mathbb{E}[|\mathbb{F}_i| \mid \sigma_i(S_{sess})]} \sim \frac{\frac{|\Delta R|}{|\mathcal{CCF}|} + \frac{|\Delta R|}{|S_{sess}(i)|} \cdot q^{|S_{sess}(i)|}}{1 + \frac{|\Delta R|}{|\mathcal{CCF}|}}. \tag{68}$$

This can be simplified as follows:

$$\frac{\mathbb{E}[|\Delta\mathbb{F}_i| \mid \sigma_i(S_{sess})]}{\mathbb{E}[|\mathbb{F}_i| \mid \sigma_i(S_{sess})]} \sim \frac{\frac{|\Delta R|}{|\mathcal{CCF}|} + \frac{|\Delta R|}{|S_{sess}(i)|} \cdot q^{|S_{sess}(i)|}}{1 + \frac{|\Delta R|}{|\mathcal{CCF}|}} = \frac{|\Delta R| \cdot \left(1 + \frac{|\mathcal{CCF}|}{|S_{sess}(i)|} q^{|S_{sess}(i)|}\right)}{|\mathcal{CCF}| + |\Delta R|}. \tag{69}$$

This can be upper bounded as follows, by distributing in the numerator and upper bounding $|\Delta R|$ by $|\Delta R| + |\mathcal{CCF}|$ in the numerator of the resulting first term:

$$\frac{|\Delta R| \cdot \left(1 + \frac{|\mathcal{CCF}|}{|S_{sess}(i)|} q^{|S_{sess}(i)|}\right)}{|\mathcal{CCF}| + |\Delta R|} \leq 1 + \frac{|\Delta R| \cdot |\mathcal{CCF}| q^{|S_{sess}(i)|}}{|S_{sess}(i)|(|\mathcal{CCF}| + |\Delta R|)}. \tag{70}$$

We can further upper bound by noticing that $|\mathcal{CCF}| + |\Delta R| \geq |\mathcal{CCF}|$, so

$$\frac{\mathbb{E}[|\Delta \mathbb{F}_i| \mid \sigma_i(S_{sess})]}{\mathbb{E}[|\mathbb{F}_i| \mid \sigma_i(S_{sess})]} \leq 1 + \frac{|\Delta R|}{|S_{sess}(i)|}. \tag{71}$$

Now, by our inductive hypothesis, we know that $|\Delta R|_i = o(\mu_{i-1}(S_{sess})^{0.66})$, and by concentration, we know that $|S_{sess}(i)| = \Theta(\mu_{i-1}(S_{sess}))$. Thus, we have

$$\frac{\mathbb{E}[|\Delta \mathbb{F}_i| \mid \sigma_i]}{\mathbb{E}[|\mathbb{F}_i| \mid \sigma_i]} \leq 1 + \frac{|\Delta R|}{|S_{sess}(i)|} = 1 + o(\mu_{i-1}(S_{sess})^{-(1-0.66)}) = O(1). \tag{72}$$

Thus,

$$\mathbb{E}[|\Delta \mathbb{F}(S_{sess}, \tilde{S}, i, i)| \mid \sigma_i(S_{sess})] = O(\mathbb{E}[|\mathbb{F}_i(S_{sess})| \mid \sigma_i(S_{sess})]). \tag{73}$$

Now, remember that our goal is to show that $|\Delta \mathbb{F}(S_{sess}, \tilde{S}, i, i)| = O(|\mathbb{F}_i(S_{sess})|)$ with high probability, conditioned on $\sigma_i(S_{sess})$. This follows from the expectation bound above and the fact that the size of the frontier in both clocks is binomially distributed, so that standard concentration bounds apply. This results in the following:

$$p_e |\Delta \mathbb{F}_i| = O(p_e |\mathbb{F}_i|) \tag{74}$$

with conditional probability at least $1 - e^{-\Omega((np)^i)}$.

**Upper bounding (40) by $o(p_e|R(S_{sess}, i)|)$:**    To upper bound (40), we note that

$$\sum_{j=0}^{i} |S_{sess}(j)| = |R(S_{sess}, i)|, \tag{75}$$

and an analogous identity holds with $\tilde{S}$ in place of $S_{sess}$. Moreover,

$$\left| R(S_{sess}, i) - R(\tilde{S}, i) \right| = |\Delta R(S_{sess}, \tilde{S}, i, i)|. \tag{76}$$

Thus, we have

$$(40) = p_e |\Delta R(S_{sess}, \tilde{S}, i, i)| \leq p_e f_1(n, i), \tag{77}$$

where the inequality is by the inductive hypothesis. We want this to be $o(p_e \cdot |R(S_{sess}, i)|)$, which means that we want $|\Delta R(S_{sess}, \tilde{S}, i, i)| = o(|R(S_{sess}, i)|)$. This follows from the inductive hypothesis. In particular, we know that $|R(S_{sess}, i)| \geq |S_{sess}(i)|$, since $S_{sess}(i) \subset R(S_{sess}, i)$. Furthermore, we have by the inductive hypothesis that $|\Delta R(S_{sess}, \tilde{S}, i, i)| = o(\mu_{i-1}(S_{sess})^{0.66}) = o(|S_{sess}(i)|^{0.66})$. Thus, we have

$$p_e |\Delta R(S_{sess}, \tilde{S}, i, i)| = o(p_e |R(S_{sess}, i)|), \tag{78}$$

with (conditional) probability 1, as desired.

**Upper bounding (41) by $\sum_{v \in \mathbb{F}_i(S_{sess})} Q(i, S_{sess}, v) p \mu_{i-1}^{2/3}(S_{sess}(i))$:**    To upper bound (41), we apply the triangle inequality and extend both sums to $v$ in $\mathbb{F}_i(S_{sess}) \cup \mathbb{F}_i(\tilde{S})$. This results in the following upper bound:

$$(41) \leq \sum_{v \in \mathbb{F}_i(S_{sess}) \cup \mathbb{F}_i(\tilde{S})} |Q(i, S_{sess}, v) - Q(i, \tilde{S}, v)|. \tag{79}$$

To proceed, we will upper bound the number of nonzero terms in (79). Each nonzero term can be upper bounded by 1, since $Q(i, S_{sess}, v), Q(i, \tilde{S}, v)$ are both probabilities. We will show that the number of nonzero terms is at most $O(|\mathbb{F}_i(S_{sess})| p \cdot \mu_{i-1}^{2/3}(S_{sess}(i)))$ with high probability.

We write

$$Q(i, S_{sess}, v) - Q(i, \tilde{S}, v) \tag{80}$$

$$= p_e + (1 - p_e)(1 - (1 - p_n)^{\deg_{S_{sess}(i)}(v)}) - p_e - (1 - p_e)(1 - (1 - p_n)^{\deg_{\tilde{S}_i}(v)}) \tag{81}$$

$$= (1 - p_e)((1 - p_n)^{\deg_{\tilde{S}_i}(v)} - (1 - p_n)^{\deg_{S_{sess}(i)}(v)}). \tag{82}$$

Thus, a term in the sum (79) is nonzero if and only if $\deg_{S_{sess}(i)}(v) \neq \deg_{\tilde{S}_i}(v)$. This happens if and only if $v$ has at least one edge to some vertex in $\tilde{S}_i \triangle S_{sess}(i)$. Thus, our task reduces to figuring out how many vertices $v$ there are that connect to some element of $\tilde{S}_i \triangle S_{sess}(i)$. The expected number of such vertices is

$$|\mathbb{F}_i(S_{sess}) \cup \mathbb{F}_i(\tilde{S})| \cdot (1 - q^{|\tilde{S}_i \triangle S_{sess}(i)|}). \tag{83}$$

This is an upper bound on the contribution of (41). We thus have

$$(41) \leq |\mathbb{F}_i(S_{sess}) \cup \mathbb{F}_i(\tilde{S})| \cdot (1 - q^{|\tilde{S}_i \triangle S_{sess}(i)|}). \tag{84}$$

Next, we show that $|\mathbb{F}_i(S_{sess}) \cup \mathbb{F}_i(\tilde{S})| = O(|\mathbb{F}_i(S_{sess})|)$. To do this, we apply the results from upper bounding (39). In particular,

$$|\mathbb{F}_i(S_{sess}) \cup \mathbb{F}_i(\tilde{S})| = |\mathbb{F}_i(S_{sess}) \cap \mathbb{F}_i(\tilde{S})| + |\Delta \mathbb{F}(S_{sess}, \tilde{S}, i, i)| \tag{85}$$

$$\leq |\mathbb{F}_i(S_{sess})| + |\Delta \mathbb{F}(S_{sess}, \tilde{S}, i, i)| = O(|\mathbb{F}_i(S_{sess})|). \tag{86}$$

Next, we show that $1 - q^{|\tilde{S}_i \triangle S_{sess}(i)|} = p\mu_{i-1}^{2/3}(S_{sess}(i))$. We can write

$$q^{|\tilde{S}_i \triangle S_{sess}(i)|} = (1 - p)^{|\tilde{S}_i \triangle S_{sess}(i)|} \sim e^{-p|\tilde{S}_i \triangle S_{sess}(i)|}, \tag{87}$$

provided that $p \cdot |\tilde{S}_i \triangle S_{sess}(i)| = o(1)$. Now from the inductive hypothesis, $|\tilde{S}_i \triangle S_{sess}(i)| = O(|S_{sess}(i)|^{2/3})$, and from Lemma 1, we know that $|S_{sess}(i)| = O((np)^i)$. Then we have that

$$1 - q^{|\tilde{S}_i \triangle S_{sess}(i)|} \leq 1 - e^{-O(p(np)^{2/3i})}. \tag{88}$$

In order for this second term to be $1 - o(1)$, it is sufficient to have that

$$p^{i+1} n^i = o(1). \tag{89}$$

This happens if and only if

$$p^{i+1} = o(n^{-i}) \iff p = o(n^{-\frac{i}{i+1}}). \tag{90}$$

This is guaranteed by our assumption that $p = o(n^{-\frac{T}{T+1}})$. Thus,

$$1 - q^{|\tilde{S}_i \triangle S_{sess}(i)|} \leq 1 - e^{-O(p(np)^{2/3i})} \sim p\mu_{i-1}^{2/3}(S_{sess}(i)). \tag{91}$$

We have shown that

$$(41) = O(|\mathbb{F}_i(S_{sess})|\dot{p}\mu_{i-1}^{2/3}(S_{sess}(i))). \tag{92}$$

Next, we show that $\sum_{v \in \mathbb{F}_i(S_{sess})} Q(i, S_{sess}, v) = \Omega(|\mathbb{F}_i(S_{sess})|)$. We have

$$Q(i, S_{sess}, v) \geq 1 - (1 - p_n)^{\deg_{S_{sess}(i)}(v)}. \tag{93}$$

Since the sum is over $v \in \mathbb{F}_i(S_{sess})$, this implies that $\deg_{S_{sess}(i)}(v) \geq 1$. So

$$Q(i, S_{sess}, v) \geq 1 - (1 - p_n) = p_n = \Omega(1). \tag{94}$$

Thus,

$$\sum_{v \in \mathbb{F}_i(S_{sess})} Q(i, S_{sess}, v) \geq |\mathbb{F}_i(S_{sess})| \cdot p_n = \Omega(|\mathbb{F}_i(S_{sess})|). \tag{95}$$

Thus, we have shown that

$$(41) \leq const \cdot \sum_{v \in \mathbb{F}_i(S_{sess})} Q(i, S_{sess}, v) \cdot (1 - q^{|\tilde{S}_i \triangle S_{sess}(i)|}) = const \sum_{v \in \mathbb{F}_i(S_{sess})} Q(i, S_{sess}, v) p \mu_{i-1}^{2/3}(S_{sess}), \tag{96}$$

with conditional probability at least $1 - e^{-\Omega((np)^i)}$.

**Completing the proof**   We combine (96), (74), and (78) to complete the proof. $\qquad\square$

So we have that the difference between $\mu_i(S_{sess})$ and $\mu_i(\tilde{S})$ is negligible in relation to $\mu_i(S_{sess})$.

Now, the next two propositions give the base case and inductive step of the proof of Theorem 4.

**Proposition 1** (Base case of the proof of Theorem 4)**.** *We have that, with probability 1, $|\Delta R_0| = |\Delta R(S_{sess}, \tilde{S}, 0, 0)| = 0$, and $|\tilde{S}_0 \triangle S_0| = 0$.*

*Proof.* This follows directly from the assumed initial conditions. $\qquad\square$

**Proposition 2** (Inductive step of the proof of Theorem 4)**.** *Assume that the inductive hypotheses (35) and (36) hold for $i$. Then we have the following:*

$$|\tilde{S}_{i+1} \triangle S_{sess}(i+1)| \leq |\Delta R_i| + o(1) = |\Delta R(S_{sess}, \tilde{S}, i, i)| + o(1) = o(\mu_{i-1}(S_{sess})^{2/3}) = o(\mu_i(S_{sess})^{2/3}). \tag{97}$$

*Equivalently,*

$$1 - \frac{|\tilde{S}_{i+1} \cap S_{sess}(i+1)|}{|S_{sess}(i)|} = o(\mu_i(S_{sess})^{-1/3}). \tag{98}$$

*Furthermore,*

$$|\Delta R_{i+1}| \leq |\Delta R_i| \leq o(\mu_{i-1}(S_{sess})^{2/3}) = o(\mu_i(S_{sess})^{2/3}). \tag{99}$$

*In other words, both inductive hypotheses Hypothesis 1 and Hypothesis 2 are satisfied for $i + 1$. This holds with probability at least $1 - e^{-\Omega(\mu_i(S_{sess}))}$.*

*Proof.* To prove this, we first need a few essential inequalities.

- By definition of the algorithm,

$$|\tilde{S}_{i+1}| \leq \mu_i(\tilde{S})(1 + \mu_i(\tilde{S})^{-1/3}), \tag{100}$$

  with probability 1.

- We will also need to prove an upper bound on $|\tilde{S}_{i+1}| - |S_{sess}(i+1)|$. In particular, we will show that with probability at least $1 - e^{-\Omega(\mu_i(S_{sess}))}$,

$$|\tilde{S}_{i+1}| \leq |S_{sess}(i+1)| \cdot (1 + O(\mu_i(S_{sess})^{-1/3})). \tag{101}$$

  We show this as follows. From Theorem 5,

$$\mu_i(\tilde{S}) \leq \mu_i(S_{sess})(1 + p\mu_{i-1}(S_{sess})^{2/3}),$$

with probability $\geq 1 - e^{-\Omega(\mu_i(S_{sess}))}$. This implies, via (100), that

$$|\tilde{S}_{i+1}| \leq \mu_i(S_{sess}) \cdot (1 + p\mu_{i-1}(S_{sess})^{2/3}) \cdot (1 + O(\mu_i(S_{sess})^{-1/3})).$$

By concentration of $|S_{sess}(i+1)|$, with probability at least $1 - e^{-\Omega(\mu_i(S_{sess}))}$, this is upper bounded as follows:

$$|\tilde{S}_{i+1}| \leq |S_{sess}(i+1)|(1 + O(|S_{sess}(i+1)|^{-1/2+const}))(1 + p\mu_{i-1}(S_{sess})^{2/3})(1 + O(\mu_i(S_{sess})^{-1/3})).$$

Now, we can see from (63) that this is equal to the desired upper bound. We have thus shown (101).

Now, with the preliminary inequalities proven, we proceed to prove the proposition. We split into two cases:

- $S_{sess}(i+1)$ **begins before** $\tilde{S}_{i+1}$ **(in other words, $|R(S_{sess}, i)| < |R(\tilde{S}, i)|$).**

  In this case, we will show (i) that $S_{sess}(i+1)$ must end before $\tilde{S}_{i+1}$ (i.e., that $|R(S_{sess}, i+1)| \leq |R(\tilde{S}, i+1)|$) with high probability, (ii) that

  $$\Delta R_{i+1} = o(\mu_i(S_{sess})^{.66}), \tag{102}$$

  and (iii) that

  $$|\tilde{S}_{i+1} \triangle S_{sess}(i+1)| \leq 2|\Delta R_i| + o(1) = o(\mu_i(S_{sess})^{.66}). \tag{103}$$

  To show that (i) is true, we note that because $S_{sess}(i+1)$ begins before $\tilde{S}_{i+1}$, $S_{sess}(i+1)$ consists of an initial segment $S_{obs}(j_1), S_{obs}(j_1 + 1), ..., S_{obs}(j_2)$ with total cardinality $|\Delta R_i|$, ending in a session endpoint (specifically, the one corresponding to $R(\tilde{S}, i)$), followed by a segment $S_{obs}(j_2 + 1), ..., S_{obs}(j_3)$ of total cardinality $|S_{sess}(i+1)| - |\Delta R_i|$, again ending in a session endpoint. This is true by definition of $\Delta R_i$. The second segment begins at the same point as $\tilde{S}_{i+1}$ (that is, $R(\tilde{S}, i) = \bigcup_{j=0}^{j_2+1} S_{obs}(j)$), and we know that it has cardinality

  $$|S_{sess}(i+1)| - |\Delta R_i| \leq |S_{sess}(i+1)| \leq \mu_i(S_{sess})(1 + \mu_i(S_{sess})^{-1/2+const}) \leq \mu_i(\tilde{S})(1 + \mu_i(\tilde{S})^{-1/3}), \tag{104}$$

  by concentration of $|S_{sess}(i+1)|$. The last inequality follows from the fact that $\mu_i(S_{sess}) = \Theta(\mu_i(\tilde{S}))$. Thus, the second segment of $S_{sess}(i+1)$ must be contained in $\tilde{S}_{i+1}$, by (100), by definition of the FastClock algorithm, as desired.

  This has the following implication: we can express $|\Delta R_{i+1}|$ as

  $$|\Delta R_{i+1}| = |\tilde{S}_{i+1}| - (|S_{sess}(i+1)| - |\Delta R_i|) \leq \mu_i(S_{sess})^{-1/3} + |\Delta R_i|. \tag{105}$$

  We have used (101). Since, by the inductive hypothesis, we have $|\Delta R_i| = o(\mu_{i-1}(S_{sess})^{0.66}) = o(\mu_i(S_{sess})^{.66})$, and since $\mu_i(S_{sess}) \to \infty$, this implies that

  $$|\Delta R_{i+1}| = o(\mu_i(S_{sess})^{.66}). \tag{106}$$

  Thus, we have established (ii).

  We next show (iii). We have

  $$|\tilde{S}_{i+1} \triangle S_{sess}(i+1)| = |\Delta R_i| + |\Delta R_{i+1}|, \tag{107}$$

  and by the proof of (ii) we can upper bound $|\Delta R_{i+1}|$ to get

  $$|\tilde{S}_{i+1} \triangle S_{sess}(i+1)| \leq 2|\Delta R_i| + o(1) = o(\mu_i(S_{sess})^{.66}). \tag{108}$$

  This completes the proof of (iii).

- **Or $S_{sess}(i+1)$ begins after or at the same time as $\tilde{S}_{i+1}$ (in other words, $|R(S_{sess}, i)| \geq |R(\tilde{S}, i)|$).**

  In this case, we will show (i) that

  $$|\Delta R_{i+1}| = o(\mu_i(S_{sess})^{.66}),\tag{109}$$

  and (ii) that

  $$|\tilde{S}_{i+1} \triangle S_{sess}(i+1)| \leq 2|\Delta R_i| + o(1) = o(\mu_i(S_{sess})^{.66}).\tag{110}$$

  To prove (i), we start by showing the following identity:

  $$|\tilde{S}_{i+1}| = |\Delta R_i| + |S_{sess}(i+1)| + |\Delta R_{i+1}|I_{i+1},\tag{111}$$

  where

  $$I_{i+1} = \begin{cases} 1 & S_{sess}(i+1) \text{ stops before } \tilde{S}_{i+1} \\ -1 & \text{otherwise} \end{cases}\tag{112}$$

  The identity (111) is a consequence of the following derivation, which relies on the definitions of all involved terms.

  $$\begin{aligned}
  &|\Delta R_i| + |S_{sess}(i+1)| + |\Delta R_{i+1}|I_{i+1} \\
  &= \sum_{k=0}^{i} |S_{sess}(k)| - \sum_{k=0}^{i} |\tilde{S}_k| + |S_{sess}(i+1)| + \left| \sum_{k=0}^{i+1} |S_{sess}(k)| - \sum_{k=0}^{i+1} |\tilde{S}_k| \right| I_{i+1} \\
  &= \sum_{k=0}^{i+1} |S_{sess}(k)| - \sum_{k=0}^{i} |\tilde{S}_k| - \left( \sum_{k=0}^{i+1} |S_{sess}(k)| - \sum_{k=0}^{i+1} |\tilde{S}_k| \right) \\
  &= |\tilde{S}_{i+1}|.
  \end{aligned}$$

  Rearranging (111) to solve for $|\Delta R_{i+1}|$, we have that

  $$\begin{aligned}
  |\Delta R_{i+1}| &= ||\tilde{S}_{i+1}| - |\Delta R_i| - |S_{sess}(i+1)|| \leq ||\tilde{S}_{i+1}| - |S_{sess}(i+1)|| + |\Delta R_i| \\
  &= ||\tilde{S}_{i+1}| - |S_{sess}(i+1)|| + o(\mu_{i-1}(S_{sess})^{2/3}) \\
  &\leq O(\mu_i(S_{sess})^{-1/3}) + o(\mu_{i-1}(S_{sess})^{2/3}) = o(\mu_i(S_{sess})^{.66}).
  \end{aligned}$$

  Here, we have used the triangle inequality and the inductive hypothesis Hypothesis 1 on $|\Delta R_i|$, followed by the inequality (101). This completes the proof of (i).

  Furthermore, this implies, by the same logic as in the previous case (108), that

  $$|\tilde{S}_{i+1} \triangle S_{sess}(i+1)| \leq 2|\Delta R_i| + o(1) = o(\mu_i(S_{sess})^{.66}),\tag{113}$$

  which verifies the inductive hypothesis Hypothesis 2 on $|\tilde{S}_{i+1} \triangle S_{sess}(i+1)|$.

The inductive hypotheses Hypotheses 1 and 2 follow directly from the above. $\qquad\square$

We can now prove the utility theorem, Theorem 4.

*Proof of Theorem 4.* Let $B_i$ denote the *bad* event that either inductive hypothesis fails to hold at step $i$. We will lower bound $\Pr[\bigcap_{i=0}^{T-1} \neg B_i]$. By the chain rule, we have

$$\Pr[\bigcap_{i=0}^{T-1} \neg B_i] = \Pr[\neg B_0] \prod_{i=1}^{T-1} \Pr[\neg B_i \mid \bigcap_{j=0}^{i-1} \neg B_j].\tag{114}$$

From Proposition 2, Proposition 1, and Lemma 1, this is lower bounded by

$$\prod_{i=1}^{T-1}(1 - e^{-D\cdot(np)^{i+1}}) = \exp\left(\sum_{i=1}^{T-1}\log\left(1 - e^{-D(np)^{i+1}}\right)\right) = \exp\left(-\sum_{i=1}^{T-1}e^{-D(np)^{i+1}}\cdot(1 + o(1))\right)$$
$$= 1 - e^{-\Omega(np)}.$$

Now, the event that none of the bad events hold implies the claim, which completes the proof. $\square$

With Theorem 4 in hand, we can prove the main result, Theorem 1.

*Proof of Theorem 1.* Let us recall the definition of $d_{S_{obs}}(C, \hat{C})$. We have

$$d_{S_{obs}}(C, \hat{C}) = \frac{1}{\binom{n}{2}}\sum_{i<j}\text{Dis}_{C,\hat{C}}(i,j). \tag{115}$$

What we need is an upper bound on this quantity in terms of the error term $f(n) = (np)^{-1/3}$ in Theorem 4. To this end, we partition the sum according to vertex membership in clock intervals as follows:

$$\binom{n}{2}d_{S_{obs}}(C, \hat{C}) = \sum_{k_1=1}^{|S|}\sum_{i<j\in S_{k_1}}\text{Dis}_{C,\hat{C}}(i,j) + \sum_{k_1=1}^{|S|}\sum_{k_2=k_1+1}^{|S|}\sum_{i\in S_{k_1}, j\in S_{k_2}}\text{Dis}_{C,\hat{C}}(i,j). \tag{116}$$

In the first sum, $i$ and $j$ are *not* ordered by $C$, because they lie in the same set in $S$. We consider the corresponding set in $\tilde{S}$. From the theorem, at least $\binom{|C_{k_1}|\cdot(1-f(n))}{2}$ vertex pairs from $S_{k_1}$ are correctly placed together in $\tilde{S}$. Furthermore, at least

$$|S_{k_1}|\cdot(1 - f(n))\cdot\sum_{k_2=k_1+1}^{|S|}(1 - f(n))|S_{k_2}| \tag{117}$$

pairs of vertices with one vertex in $S_{k_1}$ are correctly placed in different intervals. So the number of correctly ordered/unordered vertex pairs is at least

$$\sum_{k_1=1}^{|S|}\left(\frac{|S_{k_1}|^2\cdot(1 - f(n))^2}{2} + \sum_{k_2=k_1+1}^{|S|}|S_{k_1}||S_{k_2}|(1 - f(n))^2\right) \sim \binom{n}{2}\cdot(1 - f(n))^2. \tag{118}$$

Since $f(n) = o(1)$, this is asymptotically equal to $\binom{n}{2}\cdot(1 - 2f(n))$.

This completes the proof. $\square$

## B.2 Proof of Theorem 2

We analyze the worst-case running time of FastClock as follows: initialization takes $O(1)$ time. The dominant contribution to the running time is the *while* loop. Since $t_{obs}$ is initially 0 and increases by at least 1 in each iteration, the total number of iterations is at most $N$. The remaining analysis involves showing that each vertex and edge is only processed, a constant number of times, in $O(1)$ of these loop iterations, so that the running time is at most $O(N + n + m)$, as claimed.

In particular, the calculation of $\mu_t$ in every step involves a summation over all edges from currently active vertices to their uninfected neighbors, along with a calculation involving the current number of uninfected vertices (which we can keep track of using $O(1)$ calculations per iteration of the loop). A vertex is only active in a single iteration of the loop. Thus, each of these edges is only processed once in this step. The calculation of $t'_{obs}$ entails calculating a sum over elements of $S_{obs}$ that are only processed once in all of the iterations of the loop. The calculation of all of the $|S_{obs}(i)|$ can be done as a preprocessing step via an iteration over all $n$ vertices of $G$. Finally, the calculation of $\mathbb{F}_{t+1}$ entails a union over the same set of elements of $S_{obs}$ as in the

calculation of the maximum, followed by a traversal of all edges incident on elements of $\tilde{S}_{t+1}$ whose other ends connect to uninfected vertices. These operations involve processing the vertices in $\tilde{S}_{t+1}$ (which happens only in a single iteration of the loop, and, thus, with the preprocessing step of calculating the $|S_{obs}(i)|$, only a constant number of times in the entire algorithm). The edges leading to elements of $\mathbb{F}_{t+1}$ from elements of $\tilde{S}_{t+1}$ are traversed at most twice in the loop: once in the building of $\mathbb{F}_{t+1}$ and once in the next iteration in the calculation of $\mu_t$.

This implies that each vertex and edge is only processed $O(1)$ times in the entire algorithm. This leads to the claimed running time of $O(N + n + m)$, which completes the proof.

## B.3 Proof of Theorem 3

To generalize the proof of Theorem 1, we need to generalize the following auxiliary results:

1. Concentration of $|S_{sess}(i)|$. This follows from the martingale difference property, immediately. Concentration is then used in the inductive proof Proposition 2, as well as in Lemma 1, which we generalize to Lemma 2.

2. Lemma 1 on the growth of $\mu_i(S_{sess})$ and $|S_{sess}(i)|$. This follows simply from the graphon assumption and from concentration of $|S_{sess}(i)|$. The resulting generalization is as follows.

   **Lemma 2** (Growth of $\mu_i(S_{sess})$ and $|S_{sess}(i)|$). *We have that, with probability at least $1 - e^{-\Omega(n\rho_n)}$, for all $i \leq T$,*

   $$\mu_i(S_{sess}) = \Omega((np_0\rho_n)^{i+1}), O((np_1\rho_n)^{i+1}), \tag{119}$$

   *where the $\Omega, O$ are uniform in $i$. Furthermore, with probability at least $1 - e^{-\Omega(n\rho_n)}$, we have*

   $$|S_{sess}(i)| = \Omega((np_0\rho_n)^i), O((np_1\rho_n)^i) \tag{120}$$

   *for every $i$.*

   *Proof.* The structure of the proof is exactly as in that of Lemma 1. Namely, we prove this by induction on $i$. The base case follows from the graphon density assumption, along with the assumed lower bound (the assumption c)) on the probability of infection of each frontier vertex to establish the bounds on $\mu_0(S_{sess})$. The bounds on $|S_{sess}(1)|$ then follow from the assumption a). The inductive step follows in exactly the same way. □

3. Theorem 5, upper bounding the difference $|\mu_i(S_{sess}) - \mu_i(\tilde{S})|$.

   **Theorem 6** (Generalized upper bound on $|\mu_i(S_{sess}) - \mu_i(\tilde{S})|$). *Granted the inductive hypotheses Hypothesis 1 and 2, we have the following upper bound:*

   $$|\mu_i(S_{sess}) - \mu_i(\tilde{S})| \leq p_1\rho_n\mu_{i-1}(S_{sess})^{2/3}\mu_i(S_{sess}), \tag{121}$$

   *with probability $\geq 1 - e^{-\Omega(\mu_i(S_{sess}))}$.*

   *Proof.* The contribution of $p_e$ in the general case is exactly as before, since $p_e$ plays the same role. We thus ignore it in the following analysis. Recalling that $\mu_i(S_{sess})$ is an expected value, we can decompose it by linearity of expectation as follows:

   $$\mu_i(S_{sess}) = \sum_{v \in \mathbb{F}(S_{sess}, i)} \Pr[X(v, i) = 1 \mid \sigma_{i-1}(S_{sess})] \tag{122}$$

   We may upper bound $|\mu_i(S_{sess}) - \mu_i(\tilde{S})|$ as follows:

   $$|\mu_i(S_{sess}) - \mu_i(\tilde{S})| \leq \sum_{v \in \mathbb{F}(S_{sess}, i) \cup \mathbb{F}(\tilde{S}, i)} |\Pr[X(v, i) = 1 \mid \sigma_{i-1}(S_{sess})] - \Pr[X(v, i) = 1 \mid \sigma_{i-1}(\tilde{S})]|.$$

   $$\tag{123}$$

Exactly as in the proof of Theorem 5, we upper bound the sum in (123) by its number of nonzero terms. By Assumption 2, any term in this sum is nonzero only if the vertex $v$ has at least one active neighbor in $\tilde{S}_i \triangle S_{sess}(i)$. The remainder of the proof, in which we upper bound the number of such vertices, is exactly the same as in the proof of Theorem 5, with the exception that $q$ is replaced by the interval $[1 - p_1\rho_n, 1 - p_0\rho_n]$. $\qquad\square$

With these ingredients, the *statement* of Proposition 2 remains the same. Its proof changes slightly: we use Theorem 6 in place of Theorem 5, and in place of $p$ we use $p_1\rho_n$.

Propositions 2 and Lemma 2 allow us to prove a generalization of the utility theorem Theorem 4, which is as follows.

**Theorem 7** (Generalization of the FastClock utility theorem). *We have that with probability $1 - e^{-\Omega(n\rho_n)}$, for every $i \le T - 1$,*

$$|S_{sess}(i) \cap \tilde{S}_i| = |S_{sess}(i)| \cdot (1 - O(n\rho_n)^{-1/3}). \tag{124}$$

*Proof.* The proof steps are exactly the same, except that we apply the lower bounds given in Lemma 2, rather than Lemma 1. $\qquad\square$

Finally, Theorem 1 generalizes to Theorem 3, whose proof remains intact, except that we use the generalized version of the utility theorem in place of Theorem 4. This concludes the proof of Theorem 3.

## C  Examples

**Example 3** (Failure to account for multiple time scales affects downstream statistical inference). *Here we describe a simple example that demonstrates that failure to account for multiple time scales in a cascade can negatively impact the accuracy of downstream statistical inference. As this is only an example, we opt for simplicity of analysis and, thus, leave certain details to the reader. These may be filled in by standard concentration arguments.*

*Specifically, we focus on the problem of cascade doubling time prediction, studied in Cheng et al. (2014). The problem is stated as follows: given cascade observations up to/including a time $t \in \mathbb{R}$ in which $m$ vertices are infected, the task is to predict an interval $[a, b]$ such that, with probability at least $1 - \delta$, for some fixed parameter $\delta > 0$, the time of the $2m$-th infection event lies in $[a, b]$.*

*Consider a small-scale cascade model $M$ given by the discretized independent cascade model as detailed in Definition 6 on a graph $G$ with $n$ vertices, with edge transmission parameter $p_n = 1$, probability of infection from an external source $p_e = 0$, and transmission timer distributed according to a geometric distribution with success parameter $\lambda$. Consider a cascade generated by our two-scale model with small-scale model $M$ and clock $C = ([0, k-1], [k, 2k-1], [2k, 3k-1], [3k, 4k-1], ...)$, for some fixed $k \in \mathbb{N}$. Assume that $G$ is a complete binary tree and that the infection starts at the root node. Finally, assume that we have observed the cascade up to and including the midpoint of session $j$: $t = (j + 1/2)k$.*

*We first calculate the typical number of vertices infected up to and including time $t$. The number of vertices infected by the beginning of session $j$ is given by $\sum_{i=0}^{j-1} 2^i = 2^j - 1$, by the geometric sum formula. This holds with probability 1. The number of vertices infected by time $t$ is then $2^j - 1 + C \cdot 2^{j+1}$, for some specific, computable constant $C > 0$. Thus, easily, the number of infected vertices approximately doubles by time $t + k$.*

*Next, we consider what our estimate of the doubling time would be if we were to incorrectly assume that the cascade is generated by model $M$ (that is, if we were to ignore the existence of the larger time scale). To do this, we need to introduce some notation:*

- $X_j(t)$ *– the number of vertices with $j$ uninfected children at the end of timestep $t$.*

- $Y(t)$ *– the number of uninfected children of infected parents at the end of timestep $t$. We call such nodes "open slots".*

- $Z(t)$ – *the number of vertices infected in timestep $t$.*

- $W(t)$ – *the total number of vertices infected by the end of timestep $t$.*

*For any $t' > t$, we have that $Z(t')$ is approximately $\lambda \cdot Y(t'-1)$, and $Y(t')$ is exactly $Y(t'-1) + Z(t')$, because each new infection removes an open slot (the newly infected vertex) and creates two new open slots (the children of the newly infected vertex). Thus, we have that $Z(t') \approx \lambda(Y(t'-2) + Z(t'-1))$, so that*

$$Z(t') \approx \lambda Y(t'-1) = \lambda(Y(t'-2) + Z(t'-1)) \tag{125}$$

$$= \lambda(Y(t) + Z(t+1) + Z(t+2) + ... + Z(t'-1)) \tag{126}$$

$$= \lambda(Y(t) + Z(t+1) + ... + Z(t'-2)) + \lambda Z(t'-1) \tag{127}$$

$$= (1+\lambda)Z(t'-1) = (1+\lambda)^{t'-t-1}Z(t+1). \tag{128}$$

*Next, note that $Y(t)$ can be related to $W(t)$: since each new infection increases the number of open slots by 1, we must have that $Y(t) = W(t) + 1$. So we have $Z(t') \approx (1+\lambda)^{t'-t-1}\lambda(W(t)+1)$. Then*

$$W(t') - W(t) = \sum_{i=t+1}^{t'} Z(i) \approx \lambda(W(t)+1) \sum_{i=t+1}^{t'} (1+\lambda)^{i-t-1} = \lambda(W(t)+1) \cdot \sum_{i=0}^{t'-t-1} (1+\lambda)^i$$

$$= \lambda(W(t)+1) \cdot \frac{(1+\lambda)^{t'-t}-1}{\lambda} = (W(t)+1) \cdot ((1+\lambda)^{t'-t}-1).$$

*Then, in order for $t'$ to be the doubling time for $t$, we must have that $((1+\lambda)^{t'-t}-1) \geq 1$. It is then easily seen that the required $t'$ is constant with respect to $k$, so that we substantially underestimate the doubling time when $k$ is large. Thus, failure to account for the larger time scale in this setting leads to substantial and, in this setting, avoidable inaccuracy.*

*More realistic empirical experiments in DiTursi et al. (2017; 2019) confirm that accounting for multiple timescales can, in practical settings, improve performance on doubling time prediction and several other downstream statistical tasks.*

**Example 4** (Infection sequences and clocks)**.** *Consider a graph $G$ with $n = 10$ vertices. A cascade may produce an observed infection sequence $S_{obs} = (\{7\}, \{4\}, \{1,3\}, \{5\}, \{2\}, \{8\}, \{\}, \{10\}, \{6,9\})$, indicating that vertex 7 is infected at time $t = 0$, 4 at time $t = 1$, 1 and 3 at time 2, etc. If the cascade was generated according to our multiscale model with session end points at $t = 1, 3, 4, 8$, then this would induce the following session infection sequence: $S_{sess} = (\{4,7\}, \{1,3,5\}, \{2\}, \{6,8,9,10\})$. The clock that induced this session infection sequence is given by the session end points. As a partition into subintervals, it is encoded by $([0,1], [2,3], [4,4], [5,8])$.*

**Example 5** (Frontier sets of an infection sequence)**.** *Consider the graph depicted in Figure 3, where sets of vertices are arranged from left to right according to the example infection sequence $S_{sess} = (\{1\}, \{3,5\}, \{2,4\}, \{6\})$. The frontier sets corresponding to $S_{sess}$ on this graph are*

$$\mathbb{F}_0(S_{sess}) = \{3,4,5,6\}, \tag{129}$$

$$\mathbb{F}_1(S_{sess}) = \{2,4,7\}, \tag{130}$$

$$\mathbb{F}_2(S_{sess}) = \{6,7\}, \tag{131}$$

$$\mathbb{F}_3(S_{sess}) = \emptyset. \tag{132}$$

## D   Empirical results on synthetic and real graphs

In this section, we give our complete set of empirical results, continuing the material in Section 3.4.

**Experiments on Erdős–Rényi graphs.**   We first experiment with Erdős–Rényi to confirm the theoretical behavior of our estimator and compare its running time and quality to the DP baseline. We report the results

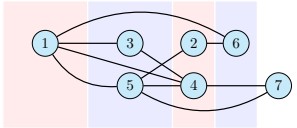

Figure 3: The network for the frontier example, Example 5. Distinct shaded regions from left to right denote distinct infection sets from the sequence $S_{sess}$ in the example.

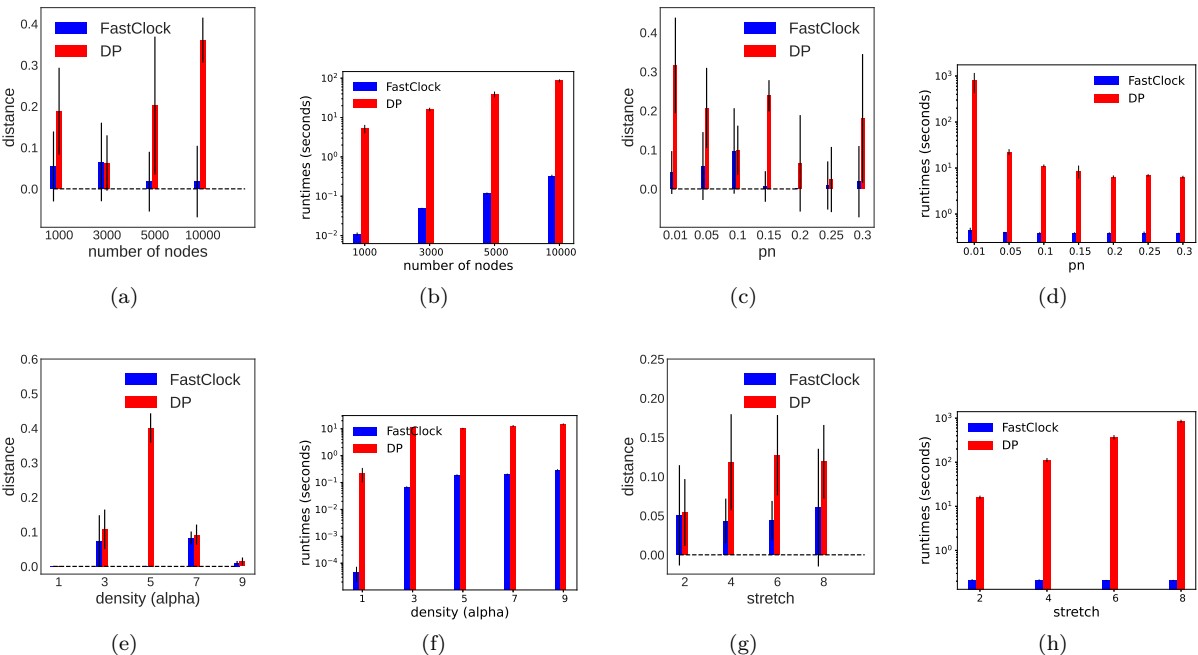

Figure 4: Comparison of the distance and runtime of the estimated clocks by *FastClock* and the baseline DP from DiTursi et al. (2017) on Erdős–Rényi graphs (default parameters for all experiments: $p_n = 0.1$, $p_e = 10^{-7}$, $n = 3000$, $p = n^{-1/3}$, stretch $l = 2$ unless varying in the specific experiment). (a),(b): Varying graph size. (c),(d): Varying infection probability $p_n$. (e),(f): Varying graph density $p = n^{-1/\alpha}$. (g),(h): Varying stretch.

in Figure 4. With increasing graph size *FastClock*'s distance from the ground truth clock diminishes (as expected based on Theorem 1), while that of DP increases (Fig. 4(a)). Note that DP optimizes a proxy to the cascade likelihood and in our experiments tend to associate too many early timesteps with early sessions, which for large graph sizes results in incorrect recovery of the ground truth clock. Similarly, *FastClock*'s estimate quality is better than that of DP for varying on $p_n$ (Fig. 4(c)), graph density (Fig. 4(e)) and stretch factor for the cascades (Fig. 4(g)), with distance from ground truth close to 0 for regimes aligned with the key assumptions we make for our main results (Assumption 1 or, more generally, 2). In addition to superior accuracy, *FastClock*'s running time scales linearly with the graph size and is orders of magnitude smaller than that of DP for sufficiently large instances (Figs. 4(b), 4(d), 4(f), 4(h)).

**Experiments on Stochastic Block Model (SBM) graphs.** We would also like to understand the behavior of our estimator on graphs with communities where the cascade may cross community boundaries. To this end, we experiment with SBM graphs varying the inter-block connectivity and virality ($p_n$) of the cascades and report results in Fig. 5. As the cross-block connectivity increases and approaches that within blocks (i.e. the graph structure approaches that of an ER graph) *FastClock*'s quality improves and is significantly better than that of DP (Fig. 5(a)). When, however, the transmission probability $p_n$ is high, coupled with sparse inter-block connectivity, *FastClock*'s estimation quality deteriorates beyond that of DP (Fig. 5(c)). This behavior is due to the relatively large variance of $\mu_t$ when the cascade crosses a sparse cut

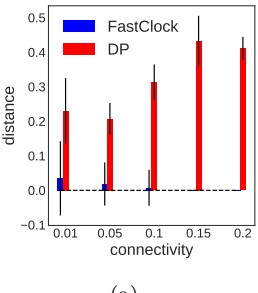 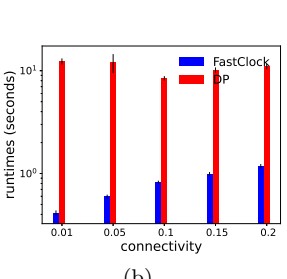 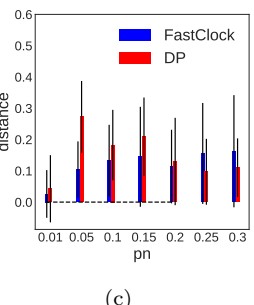 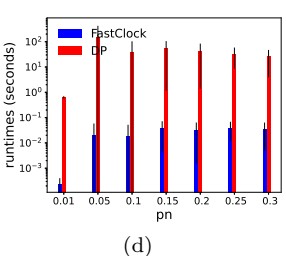

|  |  |  |  |
|:---:|:---:|:---:|:---:|
| (a) | (b) | (c) | (d) |

Figure 5: Comparison of the distance and runtime of the estimated clocks by *FastClock* and the baseline DP from DiTursi et al. (2017) on Stochastic Block Model graphs (default parameters: $n = 5000$, two blocks/communities of sizes $n/\sqrt{n}$ and $n - n/\sqrt{n}$, $p_e = 10^{-7}$, stretch $l = 2$). (a),(b): Varying inter-block connectivity ($p_n = 0.1$) where a setting of 0.2 makes the graph equivalent to an Erdős–Rényi graph with $p = 0.2$. (c),(d): Varying infection probability $p_n$ (inter-block connectivity is set to 0.01).

in the graph with high probability. This challenging scenario opens an important research direction we plan to explore in future work.

### D.1 Empirical results on a real graph

Here we evaluate the performance of FastClock and the DP algorithm from DiTursi et al. (2017) via synthetic cascades on a real network. The graph in question, harvested by Bogdanov et al. (2013), is a Twitter subnetwork consisting of 3000 nodes, with edges representing follower-followee relationships. The network was constructed by a breadth-first search starting from a set of seed nodes consisting of the authors of that work and their labmates. We generated cascades on this network with a single random initially infected node and varied $p_n$ from 0.01 to 0.15. We plotted the accuracy and running time (Figure 6) of the two algorithms versus $p_n$, averaged over 50 trials. For all values of $p_n$, the running time of FastClock is substantially smaller than that of the DP algorithm. Regarding accuracy, for smaller $p_n$, FastClock's average distance is smaller than or equal to that of DP. The accuracy of FastClock decays past approximately $p_n = 0.15$ but remains below an average distance of 0.07. We note that these larger values for $p_n$ may not be practically relevant. E.g., in Kempe et al. (2003), values of $p_n$ between 0.01 and 0.1 (and, in general, varying inversely in proportion to the degrees of nodes) were considered relevant.

## E  Discussion of knowledge of small-scale cascade model parameters

Our algorithm relies on knowledge of the parameters of the small-scale cascade model. Here we discuss this aspect further. Our main messages are as follows:

1. Joint estimation of small-scale model parameters and the clock from a sample cascade is, without any assumptions, information-theoretically impossible. This is to be expected: the small-scale model parameters determine the local temporal dynamics of the process, and large-scale effects can distort these local dynamics.

   This makes the issue subtle and motivates work that is beyond the scope of the present paper.

2. In many use cases, small-scale model parameters may be estimated from data beyond sample cascades. We give an example setting where this is plausible. In other settings, with certain assumptions, joint estimation from a sample cascade is possible. The main message here is that estimation of small-scale model parameters is of interest, but a full exploration of practical methods is beyond the scope of this paper. It is appropriate and plausible to regard these cascade parameters as being given to us.

3. We can show that our algorithm is robust to random model parameter uncertainty.

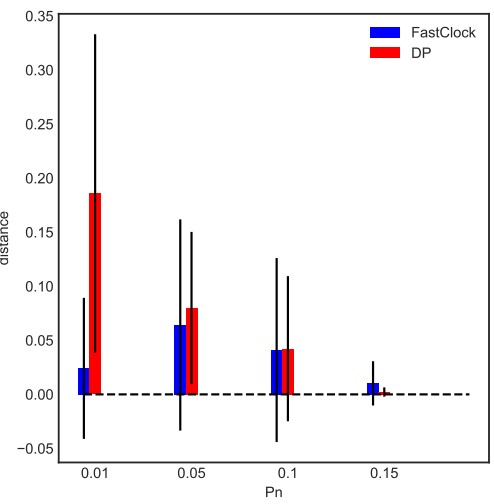 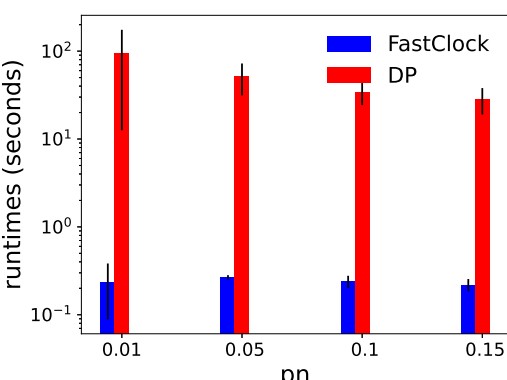

Figure 6: 50 Synthetic cascades generated on a real-world Twitter graph of about 3000 nodes. a and b) distance and runtime over $p_n$. DP exhibits higher accuracy at higher $p_n$ and FastClock does better at lower $p_n$. FastClock is much faster than DP, but loses a small amount of accuracy.

### E.1 Estimation of transmission probabilities from non-cascade data

In some situations, cascade model parameters (i.e., transmission probabilities for individual edges) are determined by observable pairwise node features and, hence, do not require cascade observations and ground truth clock information for estimation. In other words, in these scenarios, it is plausible and appropriate to think of the cascade model parameters as being already determined and available to us when we observe sample cascades and estimate clocks.

More concretely, nodes $v$ may be endowed with feature vectors $\vec{f_v} \in \mathbb{R}^d$, for some dimension $d$, and the single-timestep transmission probability across an edge $(v, w)$ may be stipulated to depend deterministically on $\vec{f_v}$ and $\vec{f_w}$, in a way to be determined. I.e., the transmission probability is stipulated to be of the form $F_\theta(\vec{f_v}, \vec{f_w})$, where $F_\theta(\cdot, \cdot)$ is a known function depending on a parameter $\theta$, which is to be estimated. This can be estimated in certain settings by experiment: for a sufficiently large number of pairs $(v, w)$ of individuals (not necessarily coming from a graph), the experimenter infects vertex $v$ and exposes $v$ to $w$. If the experimenter can reliably determine whether or not $w$ becomes infected, then this provides an estimate of the probability of transmission $F_\theta(\vec{f_v}, \vec{f_w})$, and could be used to estimate $\theta$.

### E.2 Robustness to cascade model parameter uncertainty

We can extend our theoretical guarantees for FastClock to the case where the small-scale cascade model parameters are not known exactly, but where instead each edge transmission probability is an independent sample from a fixed **unknown** distribution with known expected value and support bounded away from 0 and 1. That is, fix an arbitrary distribution $\mathcal{D}$ supported on $[A, B]$, where $0 < A \le B < 1$, with known expected value $p_n$. We stress that we do not know $A$ or $B$. For each ordered pair of vertices $(v, w)$ such that $\{v, w\}$ is an edge in the graph, $p_n((v, w)) \sim \mathcal{D}$.

The FastClock algorithm's guarantees remain the same. This is because the crucial property on which it relies is concentration of each $|S_{sess}(i)|$ around its expected value, conditioned on $\sigma_{i-1}(S_{sess})$. This conditional expectation is $\mu_i(S_{sess})$ in the setting where $\mathcal{D}$ assigns probability 1 to some fixed $p_n$ (or all of the $p_n(e)$ are deterministic). When $\mathcal{D}$ is a nontrivial distribution, the analysis is slightly more complicated: we must show that $|S_{sess}(i)|$ remains concentrated around its conditional expected value (call it $\overline{|S_{sess}(i)|}$), but this number is no longer exactly equal to $\mu_i(S_{sess})$. Under our assumptions on the random graph model, $\overline{|S_{sess}(i)|}$ is asymptotically close to $\mu_i(S_{sess})$ whenever $|S_{sess}(i-1)| \to \infty$. This is the case when $i > 1$. Thus, the only

trouble comes when $i = 1$ and the seed set $S_{sess}(0)$ has small cardinality ($\Theta(1)$ with respect to $n$). In this case, with high probability, every vertex in $\mathbb{F}_0(S_{sess})$ has exactly one neighbor in $S_{sess}(0)$ (unless the graph is extremely dense). For each vertex $v \in \mathbb{F}_0(S_{sess})$, let $e(v)$ denote this unique edge from $v$ to $S_{sess}(0)$. Then $|S_{sess}(1)|$ is asymptotically equivalent to $\mu_1(S_{sess})$ provided that

$$\sum_{v \in \mathbb{F}_0(S_{sess})} (1 - p_n(e(v))) \sim \sum_{v \in \mathbb{F}_0(S_{sess})} (1 - p_n). \tag{133}$$

This is trivially the case because $\mathbb{E}[p_n(e(v))] = p_n$ and because the $p_n(e(v))$, for different $v$, are independent.

Thus, our algorithm's accuracy and running time guarantees are robust to random uncertainty in model parameters.

# F    Estimation of $S_{sess}(0)$

Here we describe how $|S_{sess}(0)|$ might be estimated under various assumptions.

In some application scenarios, we are free to choose cascade seeds: e.g., in influence maximization. In this case, the problem is trivial.

Alternatively, if we assume that the seeds are uniformly randomly chosen, then with high probability they form an independent set (if the graph is sparse enough). We can estimate $S_{sess}(0)$ to be $S'_0 = \bigcup_{j=0}^{k_*} S_{obs}(j)$, where $k_* = \max\{k : \cup_{j=0}^{k} S_{obs}(j) \text{ is an independent set in G}\}$. This yields the correct result with high probability if $p_e$ is small enough (which we assume in our theorems).

Without any assumptions, any algorithm must be given $S_{sess}(0)$, because, e.g., it is impossible to distinguish between a cascade $C$ with $T$ sessions versus zero sessions of a cascade whose seed set is the final state of $C$.

# G    Behavior of FastClock under deviations from the modeling assumption

Here we consider deviations from one of the modeling assumptions and their consequences for FastClock. Specifically, we consider what happens when, within any given session, a small set of vertices that become infected also can immediately begin the process of infecting a subset of their neighbors.

To examine this scenario, we consider a relaxation of our model, wherein, in each session $j$, a subset of at most $k_j$ vertices may be "non-compliant", in the sense that they become active immediately. Let us call the non-compliant set for session $j$ $NC_j$. In this case, the total number of vertices in the infection trees of these vertices is at most $\sum_{v \in NC_j} \deg(v)$, which is, with high probability, at most $O(k_j np)$. Provided that $k_j \ll (np)^j$, this does not affect the performance guarantee on FastClock.

In an extreme scenario where *all* vertices are non-compliant, we can modify the FastClock algorithm as follows: starting at the beginning of each session, we keep a running total of the conditional expected number of vertices infected since the start of the session. Simultaneously, we keep a running total of the number of vertices actually infected since the session start. At the first time point at which these two numbers differ by more than a certain threshold, we declare the *previous* timestep to be the end of the current session. The threshold is chosen in the same manner as in the original FastClock algorithm, to provide a certain probability of error guarantee.

# H    Real network statistics

Table 1 lists the structural statistics for large online social networks often used in empirical network science research. Such networks are well within the expected density ranges we employ in our theoretical analysis, namely their average degree is in the order (and typically higher) than a natural logarithm of the number of nodes.

| Name | Description | Type | Nodes $N$ | Edges | AVG degree | ln(N) |
|---|---|---|---|---|---|---|
| ego-Facebook | Social circles from Facebook (anonymized) | Undirected | 4,039 | 88,234 | 22 | 8 |
| ego-Gplus | Social circles from Google+ | Directed | 107,614 | 13,673,453 | 127 | 12 |
| ego-Twitter | Social circles from Twitter | Directed | 81,306 | 1,768,149 | 22 | 11 |
| soc-Epinions1 | Who-trusts-whom network of Epinions.com | Directed | 75,879 | 508,837 | 7 | 11 |
| soc-LiveJournal1 | LiveJournal online social network | Directed | 4,847,571 | 68,993,773 | 14 | 15 |
| soc-Pokec | Pokec online social network | Directed | 1,632,803 | 30,622,564 | 19 | 14 |
| soc-Slashdot0811 | Slashdot social network from November 2008 | Directed | 77,360 | 905,468 | 12 | 11 |
| soc-Slashdot0922 | Slashdot social network from February 2009 | Directed | 82,168 | 948,464 | 12 | 11 |
| wiki-Vote | Wikipedia who-votes-on-whom network | Directed | 7,115 | 103,689 | 15 | 9 |

Table 1: Statistics of some of the real-world social and information network datasets from SNAP (`http://snap.stanford.edu/data`).

