# OpenReview forum: "Fast and Accurate Spreading Process Temporal Scale Estimation"
_TMLR — Accepted by TMLR_

### Review · Reviewer_JRZX · 2022-08-31

**Summary Of Contributions:**

In this paper, the authors introduce a novel model for spreading processes on graphs, along with a fast algorithm to estimate key parameters of this model. The cornerstone of of their model is the introduction of two timescales, regulated by a \emph{clock}. Roughly speaking, the clock groups several time steps together, and "activates" newly infected nodes during the next group, which in principle allows them to start spreading along the edges of the graph. Given an observed sequence of infected nodes, the goal of the proposed algorithm is to estimate the underlying clock, or, in other words, group time steps together such that each group correspond to new unobserved "activation".

On the theoretical side, they show that their algorithm (FastClock) approaches the true clock under some martingale assumption for the spreading process and semi-sparse graphon model for the graph structure. They give an example, the Independant Cascade (IC) process.

On the practical side, the authors show that FastClock is more efficient than another dynamic programming approach and exhibit good results on synthetic Erdo-Renyi and SBM graphs and IC process.

**Requested Changes:**

- Some notation are slightly poorly chosen, most of all the letter $S$ for both the small-scale (with a hat) and large-scale steps. With only tilde and hat, it is easy to get lost between what is observed, estimated, inferred, ground-truth, at which time-scale, etc.

- the "frontier" set is absolutely crucial in the analysis and would deserve a more detailed introduction and description, maybe with an illustration ? Currently, there is also a mistake in it's definition (index $j$ not used), and the notation changes in later pages, $\mathbb{F}_t$ vs $\mathbb{F}(\cdot, t)$

- a suggestion, I think it would be clearer (and avoid repetition of proof) to first state and prove the main theorem 3, then apply it to the IC process by describing how the hypotheses are satisfied in this case (if they are?).

**Strengths And Weaknesses:**

Strength:
- the proposed model is quite general, and should allow for different cascade process.
- the proposed algorithm is simple but computationally efficient. Unless I am wrong, it is even somewhat "online", never going back on the chosen times, which is the golden standard in this kind of methods. However, it raises the question whether the results could be even more refined by going offline, and sacrificing a little bit of computational efficiency for performance results.
- the theoretical analysis is useful and non-trivial, with what seems to be a quite general framework
- the paper is generally clearly written

Weaknesses/questions:
- Some clarification could be brought about the model itself, maybe with additional remarks and/or examples.

Some parts of the model seem very permissive: eg, disregarding the theoretical analysis, the cascade model can actually be anything (def 1), not restricting the spreading to neighbors of activated nodes ? (maybe this should be written, it is a bit contradictory to what if written just above def 1). On the other hand, some parts of the model seem on the contrary a bit constrained: if I understand correctly, the clock "ticks" each time one or more nodes become "activated" ? Or, conversely, \emph{all} newly infected nodes become activated at the next ticking of clock. It feels that some if this could be relaxed: more than three node states could be easily incorporated (since the cascade model is anything), a more general process could be introduced to describe what happens when the clock ticks, etc.

The IC example is not really helpful to understand the flexibility of the proposed model, as the "clock" does not seem to play an important role. In fact, it is a bit unclear how the proposed model can faithfully modelize the example given by the authors of different activity period during the day (fast vs slow), unless the clock if very constrained and almost deterministic, which seems contrary to the basis of the computations in FastClock which lie of the the computation of an expectation. It would be greatly helpful to describe a model like this, and what the computation look like in this case.

More pictural illustrations of various spreading processes could be very helpful.

- the clock estimation problem seems to bear many ressemblance to multiple change-point detection, where the goal is to identify multiple time points where the distribution of a time series changes, which is not mentioned by the authors. It would be good to comment on the link between the two literatures and their algorithms. Especially, the distinction between online/offline algorithms is crucial in the change-point literature. FastClock may bear ressemblance with existing algorithms for online change-point detection.

---

### Review · Reviewer_Q2i7 · 2022-09-04

**Summary Of Contributions:**

The authors consider the problem of estimating time scales in spreading processes on networks, such as cascades of information diffusion. Such cascades are often modeled at short time scales assuming that nodes are always in a position to infect other nodes. This assumption is unrealistic for many settings, e.g. people checking social media in "sessions" while they are not actively at work. The authors instead propose a model parameterized by a *clock* that activates the cascade model and determines which nodes are active and able to infect others.

The main contributions I observe are as follows:
- They propose a fast (linear-time) algorithm called FastClock to estimate the clock given the node infection times.
- They provide approximation guarantees for the FastClock algorithm, showing that the clock estimation error decreases to 0 at a polynomial rate as a function of the average degree of the network.
- The FastClock algorithm is both faster and more accurate than a prior approach from DiTursi et al. (2017) on simulation experiments. This contribution is unfortunately buried in the appendices currently.

**Broader Impact Concerns:**

No broader impact statement is present, and I do not think one is necessary.

**Requested Changes:**

I have only one major change to request: Move some of the empirical results currently in Appendix D into the main paper. For example, Figure 2(a)-(d) would be sufficient to provide the reader with at least some empirical evidence of the claimed improvement in runtime and accuracy compared to DiTursi et al. (2017). The bulk of the empirical results can stay in the appendix. If more room is needed in the main paper, I suggest moving some of Section 3.4 describing the more general theoretical results into the appendices.

Minor issues:
- Consider switching to a log scale or finding some other way to present the comparison between FastClock and DP in Figure 2. The blue bars for FastClock are not visible in many of the plots.
- Definition 3: number sessions -> number *of* sessions
- Definition 6: node v become active -> node v *becomes* active


**Strengths And Weaknesses:**

Strengths:
- Proposed FastClock algorithm run time scales linearly with the number of sessions and the number of edges in the graph when the discretized independent cascade (IC) model is used as the small-scale cascade model.
- The FastClock algorithm can be used with any small-scale cascade model, not just the discretized IC model, and will be efficient as long as the small-scale model allows expected number of infected nodes to be calculated efficiently.
- The authors provide sufficient conditions for the small-scale model and random graph model in order for their guarantees to generalize beyond the discretized IC model on an Erdős-Rényi graph.
- The paper is very well written. The authors have motivated their problem formulation in an intuitive manner and provided very readable summaries of their assumptions and key theoretical results.

Weaknesses:
- The authors assume that the parameters of the small-scale cascade model are known. The authors specify good reasons why they make this assumption in Appendix E. For example, it may not be possible to both estimate the small-scale model parameters and the clock from a cascade because they interact with each other in the generative model, which then makes it impossible to evaluate the clock estimation error. However, I argue that it could still be possible to evaluate accuracy on some sort of downstream task, such as the cascade doubling time prediction that the authors have mentioned.
- All empirical results are buried in Appendix D. This is highly disappointing given that the authors claim the empirical results as one of their contributions. This can be fixed without too much effort though--see my requested changes.

---

### Review · Reviewer_9pKo · 2022-09-16

**Summary Of Contributions:**

The paper studies spreading processes on graphs and formulates a model that incorporates multiple temporal scales of cascade behavior. The goal is to allow the models to capture both small time scale cascade behaviour but also intermittent longer time-scale behaviour. The model introduces a “clock” parameter that encodes when sessions of cascade activity start. The sessions are governed by a small-scale cascade model. Parameter estimation for the model leads to the task of clock estimation.
The paper makes the following contributions:
(1) The paper formulates a distortion measure and then proposes an estimation algorithm that has linear complexity with respect to the size of the input.
(2) Theoretical results are provided to demonstrate that the proposed algorithm is statistically accurate provided reasonable conditions are satisfied by the spreading process and the graph. The initial theoretical results in the paper focus on the special case where the incorporated model is a discretized independent cascade process and the graph is Erdös-Rényi, with constraints on the edge probability and its relationship to the number of sessions. Subsequently, a considerably more general result is provided, with the conditions being extended to sparse graphon models for the graph and to cascade processes that satisfy a martingale difference property.
(3)  Experimental results (in an appendix) demonstrate that the proposed method outperforms the state-of-the-art competitor which is based on dynamic programming, at least for synthetic settings where the cascade behaviour and graphs adhere to the modelling assumptions.


**Broader Impact Concerns:**

In my view there are no concerns.

**Requested Changes:**

(1)	Justification and utility of the proposed model
The presented model corresponds to few models previously presented in the literature and, in this paper, it is not justified or supported by any data where this form of cascade behaviour is observed. If the model is exactly the same as the single clock model in DiTursi et al. (2017), then I would very strongly recommend that the authors acknowledge this in the introduction. Buried in Appendix C is the statement “More realistic empirical experiments in DiTursi et al. (2017; 2019)…”, but the submitted paper does not state explicitly that the models are the same, and I do not think this would be obvious to a reader without a careful reading of DiTursi et al. DiTursi et al. is not even cited in the introduction. The phrasing in the related work regarding the DiTursi et al. paper is cryptic: “the authors formulated a version of the problem of clock recovery from cascade data”. The contribution of this paper is not undermined by acknowledging that the model formulation appeared in an earlier work; in my view it is the formulation of the estimation task and identification of a suitable distortion metric that are the key foundational advances of this work.

The paper makes attempts to justify and motivate the assumptions, but the arguments are not particularly convincing. Many of the arguments are unsupported by sufficient citation to related literature (e.g., the introduction, which includes a section on “Motivating applications”, cites only two supporting papers, and it is not clear that these provide direct support for a session model).  In particular:

(a)	Some of the motivating examples involve synchronized activity sessions, but at the same time, these need to be somewhat unpredictable, requiring to be learned from the data. Otherwise, we could simply say that a morning session lasts from 8am until 12pm and forego any estimation. In a real-world setting, it’s challenging to envisage settings where both of these conditions arise. Perhaps the authors can provide a more practical example, going beyond a simple appeal to a social network setting.

(b)	The requirement that a user becomes infected in one session but then only becomes active in the next session also seems very restrictive. The paper argues that this is “justified in scenarios where nodes are not immediately infectious when they become infected”. It’s challenging to see a setting where all infected users would be delayed for just the right amount of time to stay uninfectious for the current session, but then all be ready to infect in the following session. I think a clearer, detailed and concrete example would help me understand the applicability of the model. Data that shows evidence of session-based cascades of this kind would be even more compelling.

(c)	Beyond providing examples in support of the adopted model, it would be very helpful if there could be some further discussion of the applicability of the proposed model and algorithm if some of the modelling assumptions are not fully met. For example, if a small fraction of users do become infectious immediately, would this lead to the algorithm and model failing completely, or would it introduce small errors. This would provide a better sense of whether the model can be applied in a practical setting even if it is just a reasonable approximation to the observed behaviour.  Appendix E does an excellent job of this for the small-scale model, but a similar type of discussion regarding the large-scale (session) model would be a very welcome addition.

(d)	In the appendix, the paper provides examples of networks that display density characteristics similar to those of the graph models used to establish the theoretical results in the paper, but there is no attempt to perform simulations of cascades on those networks. This is an opportunity to explore performance when there is a departure from the assumptions used to derive the theoretical results. While I appreciate that this is a paper that is focused on theoretical developments, and I am not requesting that the authors conduct extensive simulations, this lack of investigation of the proposed cascade model on real-world networks raises further questions in the mind of a reader. Even if the session-based cascade model itself is assumed reasonable, does the approach work reasonably well if the cascade occurs over a real-world network?

More minor issues:

(2)	The abstract claims that “our algorithm substantially outperforms the dynamic programming algorithm in terms of both running time and accuracy” and the paper states that “We empirically showed that the FastClock algorithm is superior in accuracy and running time to the current state of the art dynamic programming algorithm”. Could the authors clarify whether the synthetic data generation process completely matches the modelling assumptions of both approaches? This question could be resolved by a clear statement in the introduction that the core models in the DiTursi et al. (2017) paper and the submitted paper are the same (or different in a clearly explained way).  If the models are different, then does the data generation process unfairly favour the proposed method?

(3)	Is it possible to avoid referring to (61) in the proof of Lemma 1? Since (61) is inside a different proof, it’s challenging to know what assumptions are in place without careful checking, since they could even be introduced in the proof. It seems that (60) and (61) are simple (and brief), so I think it would be fine to repeat them.

(4)	Throughout the proof of Theorem 5, there is a running assumption (without loss of generality) that $|R(S,i)|\leq |R\tilde{S},i|$. This is
introduced in parentheses only during the upper-bounding of (38), but is then used repeatedly later. It would be helpful to introduce this assumption earlier and explain that it will be employed throughout the proof. Related to this, I found the discussion concerning $\tilde{S}_i – S_i$ after (80) challenging to follow. It wasn’t obvious to me why we couldn’t also consider edges to $S_i - \tilde{S}_i$ for (80) to be non-zero. This may just be my confusion, and it may be obvious to the authors, but it probably would be beneficial to add a sentence to explain why this is the case. If it is derived as a result of focusing on a specific case (w.l.o.g.), then it should definitely be statement explaining this.

Minor points
Figures 2 and 3 could be improved significantly by adjusting the width of the bars to improve visibility and make comparison between the two algorithms easier.
Above (37): will later be used verify -> will later be used to verify
Thus, it v is only -> Thus, v is only
Above (76): This is follows from -> This follows from


**Strengths And Weaknesses:**

Strengths
(1) Overall the paper is very well written. The theorems are clearly presented and the authors have done an excellent job of providing well-structured and well-explained proofs.
(2) For settings where the model applies, the proposed algorithm is efficient and effective.
(3) There is substantial value in the characterization of the statistical accuracy of the estimation. The results are nicely extended to sparse graphon models and relatively flexible within-session cascade models.

Weaknesses
I have one significant concern with the paper that would lead me to recommend substantial modifications prior to publication. My main concern is that the session-level model appears to be somewhat contrived. While a model-based approach can have significant advantages in terms of enabling the derivation of principled algorithms and theoretical characterization of performance, it is important to avoid specification of a model that is neither representative of real-world data nor simple enough to provide insights into general network behaviour. I detail this concern below in the requested changes.

---

### Decision · Action_Editors · 2022-10-31

**Recommendation:** Accept as is

**Comment:**

The reviewers all share the view that the paper makes a worthy technical contribution through the introduction of a new model, an algorithm for 'clock estimation' of cascade processes and its analysis. The reviewers also note that their initial comments have been adequately addressed by the authors, and feel that the paper is worthy of publication in its current state, hence my decision.

**Audience:**



**Claims And Evidence:**